# Latent Action Pretraining From Videos

**Seonghyeon Ye**[1][*][†]   **Joel Jang**[2][*][‡]
**Byeongguk Jeon**[1]   **Sejune Joo**[1]   **Jianwei Yang**[3] **Baolin Peng**[3]   **Ajay Mandlekar**[4]
**Reuben Tan**[3]   **Yu-Wei Chao**[4]   **Yuchen Lin**[5]   **Lars Liden**[3]
**Kimin Lee**[1][§]   **Jianfeng Gao**[3][§]   **Luke Zettlemoyer**[2][§]   **Dieter Fox**[2,4][§]   **Minjoon Seo**[1][§]

[1]KAIST   [2]University of Washington   [3]Microsoft Research
[4] NVIDIA   [5] Allen Institute for AI

## Abstract

We introduce Latent Action Pretraining, the first unsupervised method for pretraining Vision-Language-Action (VLA) models without ground-truth robot action labels. Existing Vision-Language-Action models require action labels typically collected by human teleoperators during pretraining, which significantly limits possible data sources and scale. In this work, we propose a method to learn from internet-scale videos that do not have robot action labels. We first train an action quantization model leveraging VQ-VAE-based objective to learn discrete latent actions between image frames, then pretrain a *latent* VLA model to predict these latent actions from observations and task descriptions, and finally finetune the VLA on small-scale robot manipulation data to map from latent to robot actions. Experimental results demonstrate that our method significantly outperforms existing techniques that train robot manipulation policies from large-scale videos. Furthermore, it outperforms the state-of-the-art VLA model trained with robotic action labels on real-world manipulation tasks that require language conditioning, generalization to unseen objects, and semantic generalization to unseen instructions. Training only on human manipulation videos also shows positive transfer, opening up the potential for leveraging web-scale data for robotics foundation models. We will open-source the model checkpoints and code at latentactionpretraining.github.io.

## 1 Introduction

Vision-Language-Action Models (VLA) for robotics (Brohan et al., 2023; Kim et al., 2024) are trained by aligning large language models with vision encoders, and then finetuning it on on diverse robot datasets (Collaboration et al., 2023); this enables generalization to novel instructions, unseen objects, and distribution shifts (Michał et al., 2024). However, diverse real-world robot datasets mostly require human teleoperation, which makes scaling difficult. Internet video data, on the other hand, offers abundant examples of human behavior and physical interactions at scale, presenting a promising approach to overcome the limitations of small, specialized robotic datasets (Yang et al., 2024c). However, it is challenging to learn from internet video data for two major challenges: first, much of the raw data on the web lacks explicit action labels; second, the data distribution from the web is fundamentally different from the embodiments and environments of typical robotic systems (McCarthy et al., 2024). We propose Latent Action Pretraining, an unsupervised approach to pretraining a robotic foundation model without the need for ground-truth robot action labels (Figure 1).

Latent Action Pretraining consists of two models that are learned sequentially, followed by a finetuning stage to map the latent actions to real robot actions. In the first pretraining stage, we use a

---

[*]Denotes equal contribution.
[†]Work done during internship at Microsoft Research.
[‡]Work done during internship at NVIDIA.
[§]Denotes equal advising.

Figure 1: **Problem Formulation.** We investigate building a generalist robotic foundation model from human motion videos without action labels.

VQ-VAE-based objective (Van Den Oord et al., 2017) to learn quantized latent actions between raw image frames. Analogous to Byte Pair Encoding (Sennrich et al., 2016) used for language modeling, this can be seen as learning to tokenize atomic actions without requiring predefined action priors (e.g., end-effector positions, joint positions). In the second stage, we perform behavior cloning by pretraining a Vision-Language Model to predict latent actions derived from the first stage based on video observations and task descriptions. Finally, we fine-tune the model on a small-scale robot manipulation dataset with robot actions to learn the mapping from the latent actions to robot actions. In this work, we refer to the resulting VLA models as LAPA.

We measure performance on diverse manipulation videos, including existing robot video datasets (without utilizing ground-truth actions) and human manipulation datasets. Our results show that the proposed method significantly outperforms baseline methods of training manipulation policies without ground-truth action labels, particularly in cross-environment and cross-embodiment scenarios. Furthermore, on real-world manipulation tasks, our method leads to a new monolithic VLA model, outperforming OPENVLA, the current state-of-the-art model Vision Language Action (VLA) model trained on a diverse mixture of datasets with ground-truth actions. These results demonstrate the effectiveness of learning unified quantized latent action representations across diverse robotic datasets featuring different embodiments (shown in Section 5.2). We further demonstrate that Latent Action Pretraining remains effective even when pretrained on *only* human manipulation video, outperforming models pretrained on Bridgev2 (Walke et al., 2023), one of the largest open-sourced robotic datasets. We observe that LAPA effectively captures environment-centric actions, including object and camera movements, which could be beneficial for downstream tasks like navigation or dynamic, non-quasistatic tasks. We expect that our method opens up the potential for building foundation models for robotics by pretraining on much larger web-scale video data.

We summarize our main contributions and findings below:

- We propose Latent Action Pretraining, an unsupervised approach to pretraining a robotic foundation model to encode robotic skills from web-scale video data.

- Experiments on simulation and real-world robot tasks show that our method not only significantly outperforms baseline methods for training robotic manipulation policies that are pretrained without using ground truth action labels, but also leads to a VLA model that outperforms the current state-of-the-art VLA model trained with ground-truth actions (by +6.22%), while achieving over 30x greater pretraining efficiency.

- We qualitatively demonstrate that it is possible to use LAPA as the action prediction model and decoder of the latent action quantization model as the world model by predicting future frames conditioned on the current observation and the latent action predicted by LAPA, effectively building a neural simulation capable of performing closed-loop evaluations entirely through neural inference.

## 2 RELATED WORK

**Vision-Language-Action Models**  Vision-Language Models (VLMs), trained on large-scale internet datasets of text, images, and videos, have shown strong capabilities in understanding and generating both text and multimodal data (Liu et al., 2023; Team, 2024; Liu et al., 2024; Abdin et al., 2024). Leveraging this, recent advancements have introduced monolithic Vision-Language-Action Models (VLAs), which extend VLMs by fine-tuning them with robotic action data for enhanced physical grounding (Brohan et al., 2023; Kim et al., 2024; Team et al., 2024; Collaboration et al.,

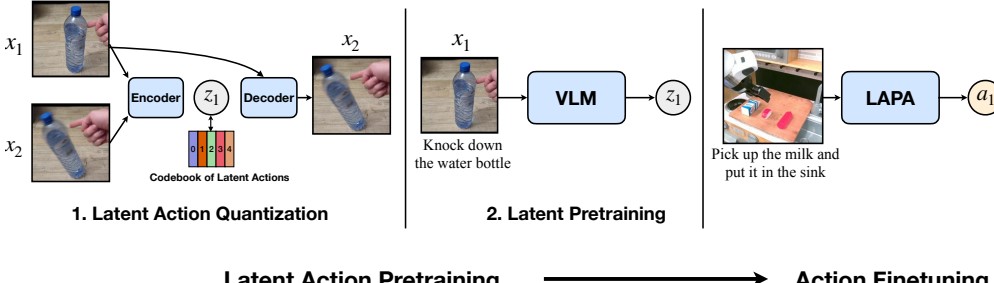

Figure 2: **Overview of Latent Action Pretraining**. (1) Latent Action Quantization: We first learn discrete latent actions in a fully unsupervised manner using the VQ-VAE objective (Detail in Figure 8). (2) Latent Pretraining: The VLM is trained to predict latent actions, essentially performing behavior cloning. After pretraining, we finetune LAPA on a small set of action-labeled trajectories to map the latent space to the end effector delta action space.

2023). Incorporating auxiliary objectives, such as visual traces (Niu et al., 2024), language reasoning paths (Michał et al., 2024), or creating conversational-style instruction datasets from robot trajectories (Li et al., 2024a), have further improved VLA performance. However, these methods remain dependent on labeled action data, limiting scalability. In contrast, our approach reduces reliance on human-teleoperated data by requiring labeled actions only for fine-tuning.

**Training Robot Policies From Videos**  Videos offer rich data for robot learning, but most lack action labels (McCarthy et al., 2024). Related work pretrains a vision encoder on egocentric human videos (Grauman et al., 2022) to improve visual representations (Nair et al., 2022; Dasari et al., 2023), or video generative models to generate future robot trajectories (Wu et al., 2024; Liang et al., 2024; He et al., 2024). Methods also extract diverse features from human videos such as interactions (Zeng et al., 2024), affordances (Bahl et al., 2023; Kannan et al., 2023; Srirama et al., 2024; Shaw et al., 2023), or visual traces (Wen et al., 2023; Bharadhwaj et al., 2024b). Some perform retargeting of human motions to robot actions to create robotic policies that involve hand pose estimators (Wang et al., 2023; Zhu et al., 2024; Shaw et al., 2023; Bharadhwaj et al., 2023; Ye et al., 2023; Qin et al., 2022) or motion capture systems (Yang et al., 2024a); these policies are usually task-specific or need aligned data human to robot data in the same environment. Finally, some train inverse dynamics models (IDMs), optical flow, or reinforcement learning models that predict actions from future state rollouts generated by world models (Du et al., 2023; Ko et al., 2024; Yang et al., 2024b; Bharadhwaj et al., 2024a) or use the IDM for active learning (Baker et al., 2022).

**Latent Actions**  Previous works have employed latent actions across diverse scenarios. GE-NIE (Bruce et al., 2024) maps user inputs (ground-truth actions) to a latent space, allowing generative models to create interactive environments. We adopt a similar latent action model but apply it to label actionless data for training a monolithic VLA to solve robotic tasks. Similarly, Edwards et al. (2018) and Schmidt & Jiang (2024) use latent actions to pretrain and fine-tune policies for video games (Cobbe et al., 2019). In contrast, we focus on learning latent actions from real-world human motions for more complex, continuous robotic tasks. Unlike other work that leverages latent actions by converting ground-truth actions into latent to capture better multimodality and task semantics Lynch et al. (2020); Jiang et al. (2023); Lee et al. (2024); Mete et al. (2024), our approach derives latent actions directly from observations, not ground-truth actions.

## 3  LAPA: LATENT ACTION PRETRAINING FOR GENERAL ACTION MODELS

Latent Action Pretraining consists of two models that are learned sequentially: Latent Action Quantization and Latent Pretraining. The overall process is illustrated in Figure 2. Note that we use the same pretraining dataset for Latent Action Quantization and Latent Pretraining.

## 3.1 Latent Action Quantization

To learn latent actions in a fully unsupervised manner, we train a latent action quantization model following Bruce et al. (2024) with a few modifications. Our latent action quantization model is an encoder-decoder architecture where the encoder takes the current frame $x_t$ and the future frame $x_{t+H}$ of a video with a fixed window size $H$ and outputs the latent action $z_t$[1]. The decoder is trained to take the latent action $z_t$ and $x_t$ and reconstruct $x_{t+H}$. Unlike Bruce et al. (2024), we use cross attention to attend $z_t$ given $x_t$ instead of additive embedding, which empirically leads to capturing more semantically meaningful latent actions. Our quantization model is a variant of C-ViViT tokenizer (Villegas et al., 2023) where the encoder includes both spatial and temporal transformer while the decoder only contains spatial transformer since our model uses only two image frames as input.

Our latent action quantization training model is based on the VQ-VAE objective (van den Oord et al., 2017), where the nearest quantized representation from the continuous embedding is retrieved from an embedding space where each embedding corresponds to a codebook. The VQ-VAE objective enables the latent action $z_t$ to be discrete tokens (codebooks), making it easy for VLMs to predict $z_t$. The latent action is represented using $s$ sequences from $|C|$ codebook vocabulary space. The sequence length $s$ is designated by the kernel size, stride and padding value of a CNN network which is used right before the vector quantization process. To avoid gradient collapse often observed in VQ-VAE, we utilize NSVQ (Vali & Bäckström, 2022) which replaces the vector quantization error to a product of original error and a normalized noise vector. We also apply stop gradient to the patch embedding of the frame $x_t$ during decoding to avoid representation collapse. Codebook replacement technique from NSVQ is applied during early training steps to maximize codebook utilization. Further model and training details are provided in Appendix A.

We utilize the encoder of our latent action quantization model as an inverse dynamics model in latent pretraining and the decoder for generating neural-based closed-loop rollouts. Unlike previous works (Bruce et al., 2024; Valevski et al., 2024), our approach trains both a world model that generates rollouts from the latent actions and a policy model that produces these latent actions through Latent Pretraining.

## 3.2 Latent Pretraining

We use the encoder of the latent action quantization model as an inverse dynamics model to label all frames $x_t$, given frame $x_{t+1}$, with latent action $z_t$. Then, we do action pretraining by using a pretrained VLM to predict the $z_t$ given the language instruction of a video clip and the current image $x_t$. Instead of using the existing language model head of the VLM, we attach a separate latent action head (a single MLP layer) of vocab size $|C|$. By default, we freeze only the vision encoder and unfreeze the language model during training. Since latent pretraining does not rely on ground truth actions, it opens the possibility of using any type of raw video paired with language instructions. Also, in contrast to traditional action granularity used in robotics (e.g. end-effector positions, joint positions, joint torques, etc.), our approach does not require any priors about the action hierarchy/granularity and is learned in an end-to-end manner simply by being optimized to best capture the 'delta' of consecutive observations in a given video dataset. We broadly refer to models having gone through latent pretraining as LAPA.

## 3.3 Action Finetuning

VLAs that are pretrained to predict latent actions are not directly executable on real-world robots since latent actions are not actual delta end-effector actions or joint actions. To map latent actions to actual robot actions, we finetune LAPA on a small set of labeled trajectories that contain ground truth actions (delta end-effector). For action prediction, we discretize the continuous action space for each dimension of the robot so that the number of data points allocated for each bin is equal following Kim et al. (2024); Brohan et al. (2023). We discard the latent action head (a single MLP layer) and

---

[1] Although Bruce et al. (2024) conditioned on multiple past observations, we exclude previous frames due to computational constraints. We leave prepending past observations as future work.

replace it with a new action head to generate ground truth actions.[2]. As with latent pretraining, we freeze the vision encoder and unfreeze all of the parameters of the underlying language model.[3]

# 4 EXPERIMENTS

In this section, we demonstrate the effectiveness of Latent Action Pretraining as a general-purpose pretaining method. Specifically, we focus on answering the following questions: **Q1.** How does LAPA perform when there are cross-task, cross-environment, and cross-embodiment gaps between pretaining and fine-tuning? **Q2.** Can LAPA learn superior priors compared to using ground-truth actions during pretraining in a multi-embodiment setting? **Q3.** Can we create a performant LAPA solely from raw human manipulation videos?

## 4.1 BENCHMARKS AND ENVIRONMENTS

We evaluate the effectiveness of LAPA on 9 different task categories in 2 different simulation environments and 3 different real-world robotic tasks. Table 3 shows an overview of the pretraining and fine-tuning dataset for each setup and Figure 9 in Appendix B visualizes the simulation benchmark and real-world setups. More details of each evaluation setup are provided in Appendix B.

**Language Table (Lynch et al., 2023)** is a simulation where a robot performs 2 DOF actions to push blocks (see Figure 9) (a)). It includes 5 subtask categories: BlocktoBlock, BlocktoAbsolute, BlocktoBlockRelative, BlocktoRelative, and Separate. During evaluation, we evaluate models for both *seen* and *unseen* scenarios, where *unseen* includes new objects (color and shape) and unseen combinations of seen objects.

**SIMPLER (Li et al., 2024b)** is a set of simulated environments for evaluating generalist robot manipulation policies. We assess our models on 4 tasks (Figure 9 (b)) using the 7 DOF WidowX robot arm. Since SIMPLER lacks fine-tuning trajectories, we collect 100 multi-task trajectories using successful rollouts from a VLA model trained on BridgeV2 data (Walke et al., 2023).

**Real-World Tabletop Manipulation** experiments used a 7 DOF Franka Emika Panda robot arm in three environments (shown in Figure 9 (c)). We utilize three pretraining data sources: Bridgev2 (Walke et al., 2023), Open-X (Collaboration et al., 2023), and Something Something v2 (Goyal et al., 2017). Following Kim et al. (2024), we finetune on three multi-instruction tasks: (1) 'Pick <object> into Sink', (2) 'Cover <object> with Towel', and (3) 'Knock <object> Over'. Each task involves 150 trajectories across 15 objects. We use a task-specific partial success criterion for evaluation, following Kim et al. (2024).

## 4.2 BASELINES

For the underlying VLM, we use the 7B Large World Model (LWM-Chat-1M) (Liu et al., 2024).

**SCRATCH** denotes the baseline model where we finetune our backbone VLM only on the downstream tasks, to quantify the gains we get from the pretraining stage.

**UNIPI (Du et al., 2023)** uses a video diffusion model during pretraining to generate video rollouts given a language instruction, which does not require any action labels during pretraining similar to our approach. For finetuning, an inverse dynamics model (IDM) is trained to extract the ground truth actions given adjacent frames.[4] We also finetune the diffusion model on the downstream task to match the target distribution.

**VPT (Baker et al., 2022)** trains an IDM on action labeled data, and uses the IDM model to extract pseudo actions on raw videos. Then, we use the pseudo actions labeled by the IDM to pretrain our backbone VLM on the pretraining data, identical to Latent Pretraining of LAPA.

---

[2]We also tried leaving the latent action head and adding additional head to decode the latent to ground-truth actions following Schmidt & Jiang (2024) However, we empirically found that re-initializing the action head resulted in superior downstream task performance, likely due to the size of the underlying policy model (7B).

[3]We leave parameter efficient fine-tuning approaches as future work for finetuning (Hu et al., 2022).

[4]We do not compare with Yang et al. (2024b) since the model is not open-sourced and Ko et al. (2024) since it is not a behavior cloning baseline.

**ACTIONVLA** denotes the baseline that uses the actual ground-truth robot action labels during pre-training with the same backbone VLM. This may be seen as the upper bound, since it utilizes the actual ground-truth labels.

**OPENVLA (Kim et al., 2024)** is a state-of-the-art VLA model that was pretrained on 970k real-world robot demonstrations from the Open X-Embodiment Dataset (Collaboration et al., 2023), mostly collected through human teleoperation. This model has a comparable number of parameters to LAPA (7B). We compare against OPENVLA for real-world robot experiments by fine-tuning the pretrained OPENVLA on our downstream tasks.

Further details of baseline models are provided in Appendix C.

### 4.3 LANGUAGE TABLE RESULTS

Table 1: **Language Table Results.** Average Success Rate (%) $\pm$ StdErr across the three different pretrain-finetune combinations from the Language Table benchmark as described in Table 3. We also note the # of trajectories used for fine-tuning next to each category.

| | In-domain (1k) | | Cross-task (7k) | | Cross-env (1k) | |
|---|---|---|---|---|---|---|
| | Seen | Unseen | Seen | Unseen | Seen | Unseen |
| SCRATCH | $15.6_{\pm9.2}$ | $15.2_{\pm8.3}$ | $27.2_{\pm13.6}$ | $22.4_{\pm11.0}$ | $15.6_{\pm9.2}$ | $15.2_{\pm8.3}$ |
| UNIPI | $22.0_{\pm12.5}$ | $13.2_{\pm7.7}$ | $20.8_{\pm12.0}$ | $16.0_{\pm9.1}$ | $13.6_{\pm8.6}$ | $12.0_{\pm7.5}$ |
| VPT | $44.0_{\pm7.5}$ | $32.8_{\pm4.6}$ | $72.0_{\pm6.8}$ | $\mathbf{60.8}_{\pm6.6}$ | $18.0_{\pm7.7}$ | $18.4_{\pm9.7}$ |
| LAPA | $\mathbf{62.0}_{\pm8.7}$ | $\mathbf{49.6}_{\pm9.5}$ | $\mathbf{73.2}_{\pm6.8}$ | $54.8_{\pm9.1}$ | $\mathbf{33.6}_{\pm12.7}$ | $\mathbf{29.6}_{\pm12.0}$ |
| ACTIONVLA | $77.0_{\pm3.5}$ | $58.8_{\pm6.6}$ | $77.0_{\pm3.5}$ | $58.8_{\pm6.6}$ | $64.8_{\pm5.2}$ | $54.0_{\pm7.0}$ |

**In-Domain Performance** First, we assess LAPA's ability to learn from a small subset of in-domain action label data by pretraining on 181k trajectories and finetuning on 1k action-labeled trajectories (0.5%). As shown in Table 1, LAPA largely outperforms SCRATCH and narrows the gap with ACTIONVLA despite not using action labels during pretraining. Additionally, LAPA surpasses UNIPI and VPT. Notably, while UNIPI handles simple tasks well, its diffusion model often generates incorrect plans for longer-horizon tasks, aligning with Du et al. (2024) (see Figure 17 of Appendix G.1). VPT, with the same backbone VLM as LAPA, outperforms UNIPI, showing the superiority of the VLA model, but still underperforms LAPA, highlighting the effectiveness of latent actions.

**Cross-Task Performance** We investigate whether LAPA's broad skills can be retained after fine-tuning on a specific task. Pretraining LAPA on 181k trajectories and finetuning on only separate tasks (7k), we evaluate all 5 task categories, similar to the in-domain setup, to assess latent pretraining's benefits for unseen tasks. When comparing LAPA and SCRATCH in Table 1 and Table 7, 8 in Appendix G.1, latent pretraining significantly benefits the separate task as well the other 4 task categories, resulting in a significant boost in both seen and unseen setups. Like before, UNIPI is constrained by its diffusion model's planning limitations, while VPT performs strongly, even surpassing ACTIONVLA in the unseen setting. This is likely due to using more labeled data (7k vs. 1k), helping the IDM generate more accurate pseudo labels.

**Cross-Environment Performance** We further investigate if Latent Action Pretraining benefits downstream performance when the pretraining and fine-tuning environments are different. We pretrain LAPA on 440k real-world trajectories, and then finetune on 1k simulation trajectories, which can be seen as testing on a setup where a real2sim gap is present (Figure 9 (a)). From Table 1, we observe that LAPA still significantly outperforms SCRATCH, showing that latent pretraining leads to positive transfer even on cross-environment setting. Notably, both UNIPI and VPT significantly underperforms LAPA, showing that learning to predict latent actions is more robust to cross-environment transfer. VPT only results in minor positive transfer, indicating that the IDM is not robust to environment shifts.

### 4.4 REAL-WORLD RESULTS

We pretrain our models on (1) Bridgev2 (Walke et al., 2023) to measure the **cross-embodiment** performance (WidowX embodiment for pretraining and Franka embodiment for finetuning) and (2) Open X-Embodiment Dataset (Collaboration et al., 2023) to measure the effect of pretraining in a

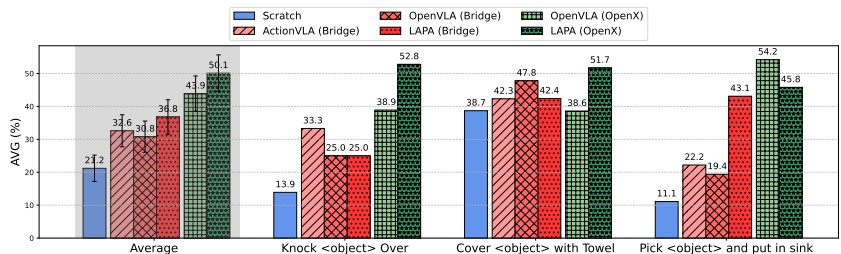

Figure 3: **Real-world Tabletop Manipulation Results.** We evaluate on a total of 54 rollouts for each model encompassing unseen object combinations, unseen objects and unseen instructions. Average success rate (%) ± StdErr are shown (detailed results provided in Appendix G.3).

Table 2: **Evaluation Results divided into eval types**. We average the success rate across the 3 tasks depending on what capability we are trying to quantify: (1) seen objects but unseen combinations, (2) unseen objects, and new instructions requiring semantic reasoning. Best is **bolded** and second best is underlined.

|  | Seen Obj. Unseen Combo | Unseen Obj. | Seen Obj. Unseen Instr. | AVG |
|---|---|---|---|---|
| SCRATCH | 18.0 | 20.3 | 25.4 | 21.2 |
| ACTIONVLA (Bridge) | 38.3 | 31.8 | 27.7 | 32.6 |
| OPENVLA (Bridge) | 35.6 | 34.6 | 22.1 | 30.8 |
| LAPA (Bridge) | 43.4 | 31.4 | 35.6 | 36.8 |
| OPENVLA (Open-X) | 46.2 | 42.1 | 43.4 | 43.9 |
| LAPA (Open-X) | **57.8** | **43.9** | **48.5** | **50.1** |
| LAPA (Human Videos) | 36.5 | 37.4 | 28.1 | 34.0 |

**multi-embodiment** setting. Figure 3 shows the average success rate across the 3 tasks where each task encompasses unseen object combination, unseen object, and unseen instruction settings. We provide detailed results depending on the generalization type in Table 2.

**Bridgev2 Pretraining** We compare models that were pretrained on the Bridgev2 dataset. Similar to previous results, all models pretrained on Bridgev2 result in significant performance enhancement compared to SCRATCH. Furthermore, by comparing LAPA which does not leverage action-labeled trajectories during pretraining with models that use action-labeled trajectories during pretraining (ACTIONVLA and OPENVLA), we observe an interesting finding: LAPA outperform VLAs that use action labeled pretraining data on average success rate of the 3 tasks, unlike previous scenarios where VLAs pretrained on the ground-truth actions were upper bounds. LAPA significantly outperforms the other models in pick-and-place tasks; given that most tasks in Bridgev2 are pick-and-place, we hypothesize that VLA models pretrained on ground truth action labels have overfitted to the WidowX action space from the Bridgev2 dataset, hampering cross-embodiment adaptability to action distribution shifts during fine-tuning. In contrast, LAPA avoids this issue by not relying on ground truth action labels during pretraining.

**Open-X Pretraining** From Figure 3, we see that VLAs pretrained on the Open-X dataset outperforms VLAs pretrained on the Bridgev2 dataset, showing that data scaling during pretraining demonstrates positive transfer for downstream tasks (Collaboration et al., 2023). This also suggests there could be significant further improvement when scaling the diversity and scale of the pretraining data, especially with large web-scale video data.

When comparing LAPA with OPENVLA, we see that LAPA significantly outperforms OPENVLA on 2 out of 3 tasks (Figure 3). Also, as shown in Table 2, LAPA (Open-X) outperforms Open-VLA (Open-X) on all types of generalization settings. This highlights LAPA's effectiveness in a multi-embodiment setting by showcasing its ability to leverage a shared latent action space during pretraining, akin to how language and image representations are utilized. In contrast, contemporary action pretraining methods may suffer from reduced positive transfer between datasets due to the variability in action representation spaces across different embodiments and datasets.

However, for pick and place task, LAPA underperforms OPENVLA. We observe that most failures of LAPA are due to early grasping. In fact, LAPA outperforms OPENVLA in reaching performance (83.33% vs 66.67%) (reaching performance for each task is provided separately in Appendix G.3).

This suggests that, although LAPA possesses stronger language conditioning and coarse-grained planning abilities, there is room for improvement in skills such as grasping. Since grasping occurs only once or twice in each trajectory, the 150 labeled trajectories may not be sufficient for LAPA to accurately predict grasp actions based on the physical characteristics of diverse objects.

## 4.5 LEARNING FROM HUMAN MANIPULATION VIDEOS

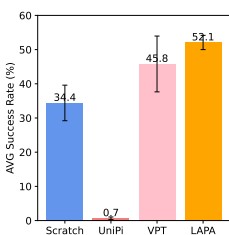

(a) SIMPLER Results

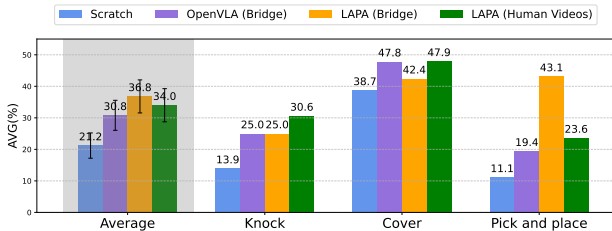

(b) Real-world Tabletop Manipulation Robot Results

Figure 4: **Pretraining from Human Video Results.** Average success rate (%) $\pm$ StdErr of LAPA and baselines pretrained on human manipulation videos where the embodiment and environment gap is extreme. We evaluate on both simulation (left) and real-world robot setup (right).

In this section, we show results when we extend Latent Action Pretraining to human manipulation videos, which aligns with the main motivation of this work. Unlike robot trajectories, human videos have two challenges: human videos do not contain action labels, and the distribution of human videos is distinct from the robot embodiment (McCarthy et al., 2024). We try to investigate whether our method as well as baseline approaches could address these challenges by pretraining on Something-Something V2 dataset (Goyal et al., 2017) which consists of 220K videos that includes human performing actions with everyday objects.

We first evaluate the performance of LAPA pretrained on human videos on SIMPLER. In addition to SCRATCH, we also compare with UNIPI and VPT pretrained with the same human video dataset. As shown in Figure 4a, LAPA outperforms SCRATCH, showing that although the distribution of the pretraining data is distinct from the deployment setup, leveraging human videos for latent action pretraining results in positive transfer. Also, LAPA performs the best performance by outperforming UNIPI and VPT, implying that Latent Action Pretraining is robust to human to robot embodiment shifts. Note that it is impossible to train ACTIONVLA because the human videos do not have any robot action labels.

We report the real-world robot experiments in Figure 4b. Surprisingly, we can see that LAPA trained with human videos outperforms OPENVLA (Bridge) on average. Despite the larger embodiment gap for LAPA (Human to robot vs. Robot to robot), it learns a better prior for robot manipulation. Also, as shown in Table 2, LAPA (Human Videos) shows good generalization performance, especially for unseen objects. We conjecture that this is because Something Something V2 dataset interacts with much diverse objects compared to Bridgev2. This result highlights the potential of raw human manipulation videos from the web compared to expensive robot manipulation data, which requires time-intensive teleoperation to collect. We expect that applying our approach on large-scale internet videos (e.g., YouTube videos) could unlock the potential for large-scale pretraining of a generalist action foundational model, similar to foundational models in NLP or Computer Vision.

## 4.6 PRETRAINING EFFICIENCY

The benefit of LAPA extends beyond downstream task performance to include pretraining efficiency. For pretraining LAPA (Open-X), the best-performing model, we use 8 H100 GPUs for 34 hours with a batch size of 128 (total of 272 H100-hours). In contrast, OPENVLA required a total of 21,500 A100-hours with a batch size of 2048. Despite being approximately 30-40 times more efficient for pretraining, LAPA still outperforms OPENVLA [5].

---

[5]We calculate based on the fact that H100 GPUs lead to 2-3 times speedup compared to A100 GPUs for training.

We believe this efficiency stems from two factors: (1) the use of the Large World Model (Liu et al., 2024) as the backbone VLM model, and (2) the coarse-grained actions of LAPA compared to conventional action pretraining. First, the training objective during LWM pretraining includes generating the next state, which corresponds to the next frame in a video. We hypothesize that this objective enables the model to implicitly understand high-level actions in a video. Notably, ACTIONVLA (Bridge), which uses LWM as the backbone, and OPENVLA (Bridge), which uses Prismatic as the backbone, are trained on the same data and objective. However, ACTIONVLA reaches optimal performance (in terms of action token accuracy) in significantly fewer epochs (3 epochs) compared to OPENVLA's 30 epochs. Second, the action space for LAPA is much smaller than that for OPEN-VLA ($8^4$ vs. $256^7$), making learning the perception-and-language to action generation problem easier to learn. For all LAPA models (BridgeV2, Open-X, Human Videos), we observe that a single epoch of training is sufficient to achieve optimal performance.

## 5 ABLATION AND ANALYSIS

### 5.1 SCALING MODEL, DATA, AND LATENT ACTION SIZE

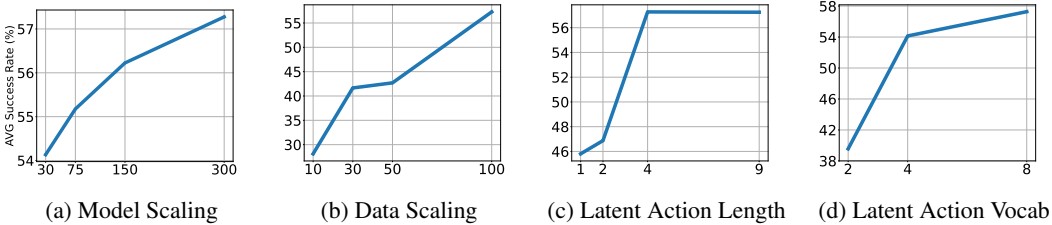

(a) Model Scaling (b) Data Scaling (c) Latent Action Length (d) Latent Action Vocab

Figure 5: **Scaling Ablation Results of LAPA**. We scale 4 dimensions of LAPA: model parameters (in millions), data size (ratio among Bridgev2), and the latent action sequence and vocabulary size, and show the downstream average success rate (%) on the SIMPLER fine-tuning tasks.

Large Language Models (LLMs) have demonstrated scaling laws (Kaplan et al., 2020), where performance improves with increases in model size, dataset size, and computational resources used for training. Similarly, we attempt to analyze whether LAPA benefits from scaling across three dimensions: latent action quantization model size, data size, and latent action representation space. For a controlled setup, we apply our method to Bridgev2 and then fine-tune it on SIMPLER except for Language Table result of Figure 5c.

As shown in Figure 5, scaling benefits LAPA across the three dimensions. Interestingly, we observe that the optimal scale of the latent action space depends on the complexity of the action dimension contained in the pretraining dataset. For example, increasing the latent action sequence length is less effective compared to increasing the vocabulary for Language Table (Figure 16). Except for Language Table, we maintain the generation space of LAPA at $8^4$ throughout all of our main experiments. These results imply that when scaling pretraining to Internet-scale videos that go beyond manipulation videos, scaling LAPA in terms of model, dataset, and latent action space could improve performance, especially to capture higher action dimensions such as whole-body control.

### 5.2 LATENT ACTION ANALYSIS

We qualitatively analyze the alignment of quantized latent actions with real continuous actions. For interpretation, we condition the current image observation $x_1$ and each latent action on the decoder of the latent action quantization model, and present the reconstructed images.

In Language Table, we observe that each latent action corresponds to a distinct movement of the robot arm, with the distribution of latent actions being well-clustered in the actual 2D action space (shown in Figure 12, 13 of Appendix E). Next, for human manipulation videos, we observe that camera viewpoints also correspond to a latent action since the viewpoint changes within a video (shown in Figure 14 of Appendix E). We also analyze the latent actions learned from the Open-X embodiment, which encompasses multiple embodiments, tasks, and environments. As shown in Figure 6, even though the embodiment and environment differ, conditioning on the same latent

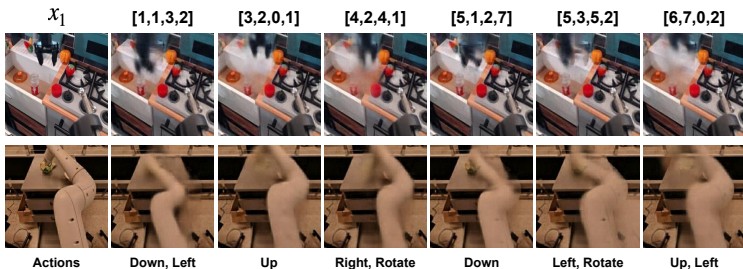

Figure 6: **Latent Action Analysis.** We condition the current observation $x_1$ and quantized latent action to the decoder of the latent action quantization model. We observe that each latent action can be mapped into a semantic action. For example, latent action [1,1,3,2] corresponds to going down and left while [3,2,0,1] corresponds to going up a little bit.

action results in a similar action in the reconstructed image. This supports our previous claim that latent actions are learned in a shared representation space, regardless of the embodiment or dataset, facilitating stronger positive transfer across diverse datasets.

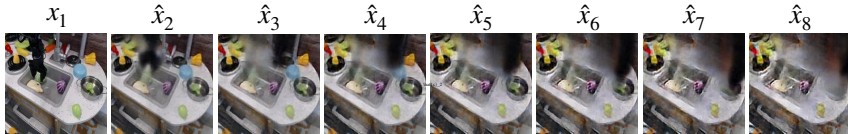

Figure 7: **Closed loop rollout of LAPA.** LAPA is conditioned on current image $x_1$ and language instruction of 'take the broccoli out of the pot'. We generate rollout images by conditioning the decoder of Latent Action Quantization Model with latent actions generated by LAPA.

We also qualitatively analyze the coarse-grained planning capability of LAPA through a closed-loop rollout. We use a LAPA model that has only undergone pretraining, without any action finetuning. Since this model generates latent actions that are not directly executable in the real world, we condition the current observation $x_1$ and the predicted latent action from LAPA with the decoder of the latent action quantization model. As shown in Figure 7, when conditioned on the current observation and the instruction to 'take the broccoli out of the pot', LAPA generates robot trajectories that successfully reaches for the broccoli, moves down to grab it, and, as the arm moves away from the pot, the broccoli disappears. This shows the potential for LAPA as a general-purpose robotic *world model*, not only predicting actions but also the outcomes of the actions. For example, this can lead to an extension of LAPA to act as a Task and Motion planning system, where it can first generate multiple plans given a natural language task instruction, choose the most optimal trajectory based on methods of quantifying the success among multiple trajectory candidates (Hwang et al., 2024; Duan et al., 2024), and perform open-loop / closed-loop inference. This can lead a paradigm where we aim to improve performance through scaling test-time compute, as with LLMs (Snell et al., 2024).

## 6 LIMITATIONS AND CONCLUSION

In this paper, we introduce Latent Action Pretraining, a scalable pretraining method for building VLAs without using ground-truth action labels. Across three benchmarks spanning both simulation and real-world robot experiments, we show that our method significantly improves transfer to downstream tasks compared to existing approaches. We also present a state-of-the-art VLA model that surpasses current models trained on 970K action-labeled trajectories. Furthermore, we demonstrate that our method can be applied purely on human manipulation videos, where explicit action information is absent, and the embodiment gap is substantial.

We still face certain limitations. First, LAPA underperforms compared to action pretraining when it comes to fine-grained motion generation tasks like grasping. We believe that increasing the latent action generation space could help address this issue. Second, similar to prior VLAs, LAPA also encounters latency challenges during real-time inference. Adopting a hierarchical architecture, where a smaller head predicts actions at a higher frequency. Lastly, while we qualitatively demonstrate that our latent action space captures camera movements (Figure 14), we have not yet explored the application of LAPA beyond manipulation videos, such as those from self-driving cars, navigation, or landscape scenes. We leave these explorations for future work.

ACKNOWLEDGMENTS

We thank Arhan Jain and Marius Memmel for helping out with the robot hardware and teleoperation setup. Also, we thank Minyoung Hwang, Jiafei Duan, Junsu Kim, and Changyeon Kim for helpful discussions and constructive feedback. This work was partly supported by Center for Advanced Urban Systems (CAUS) of Korea Advanced Institute of Science and Technology (KAIST) funded by GS E&C (40%) and the Institute of Information & Communications Technology Planning & Evaluation(IITP) grant funded by the Korea government(MSIT) (RS-2024-00397966, Development of a Cybersecurity Specialized RAG-based sLLM Model for Suppressing Gen-AI Malfunctions and Construction of a Publicly Demonstration Platform, 30%; No.RS-2022-II220264, Comprehensive Video Understanding and Generation with Knowledge-based Deep Logic Neural Network, 20%; No.RS-2021-II212068, Artificial Intelligence Innovation Hub, 10%).

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

# A    LATENT ACTION QUANTIZATION DETAILS

Figure 8: **Model architecture of our Latent Action Quantization Model.**

We show model architecture details of our latent action quantization model in Figure 8. We utilize the C-ViViT model architecture from Villegas et al. (2022) to replicate the latent action model from GENIE (Bruce et al., 2024). During the encoding process, both $x_1$ and $x_2$ are gone through a patch embedding to obtain $p_1$ and $p_2$ and gone through a spatial transformer. To convey temporal information, the representations for both outputs of spatial transformer are passed to a causal transformer (transformer with causal positional encodings) to get $e_1$ and $e_2$ continuous embeddings. We then define $d_1 = e_2 - e_1$ and discretize $d_1$ by finding the closest embedding from the codebook $z$ where the codebook size is determined as a hyperparameter.

$$z_1 = \arg\min_{z_k} |d_1 - z_k|^2 \tag{1}$$

When obtaining $e_1$ and $e_2$, we go through a CNN network. The sequence length is designated by the kernel size, stride and padding value of a CNN network. After quantization, we apply NSVQ technique before decoding.

$$\hat{d}_1 = d_1 + \frac{\|d_1 - z_1\|}{\|v\|} v \tag{2}$$

where $v \sim \mathcal{N}(0, 1)$. For decoding, we go through the following equation:

$$\hat{x}_2 = D(Attn(sg[p_1], \hat{d}_1, \hat{d}_1) \tag{3}$$

where stop gradient ($sg$) is applied to $p_1$ to avoid representation collapse and cross attention is used to attend $\hat{d}_1$ given $p_1$. Unlike the encoder, decoder $D$ only include spatial transformer. The training objective is a L2 reconstruction loss.

$$L = \|x_2 - \hat{x}_2\|_2^2 \tag{4}$$

Table 3: **Pretraining and fine-tuning dataset for each environment.** Cross-Env denotes cross-environment, Cross-Emb denotes cross-embodiment, and Multi-Emb denotes multi-embodiment. For fine-tuning, MT denotes multi-task training and MI denotes tasks with diverse multi-instructions. Category denotes the main capability we are trying to quantify.

| Environment | Category | Pretraining | | Fine-tuning | |
|---|---|---|---|---|---|
| | | Dataset | # Trajs | Dataset | # Trajs |
| LangTable | In-Domain | Sim (All 5 tasks) | 181k | 5 Tasks (MT, MI) | 1k |
| | Cross-Task | Sim (All 5 tasks) | 181k | 1 Task (MI) | 7k |
| | Cross-Env | Real (All 5 tasks) | 442k | 5 tasks (MT, MI) | 1k |
| SIMPLER | In-Domain | Bridgev2 | 60k | 4 Tasks (MT) | 100 |
| | Cross-Emb | Something v2 | 200k | 4 Tasks (MT) | 100 |
| Real-World | Cross-Emb | Bridgev2 | 60k | 3 tasks (MI) | 450 |
| | Multi-Emb | Open-X | 970k | 3 tasks (MI) | 450 |
| | Cross-Emb | Something v2 | 200k | 3 tasks (MI) | 450 |

After latent model training, we utilize the $z_1$ as the latent action label for $x_1$. The encoder can be seen as the inverse dynamics model and the decoder can be seen as the world model.

# B   DETAILS ON EXPERIMENTAL SETUP

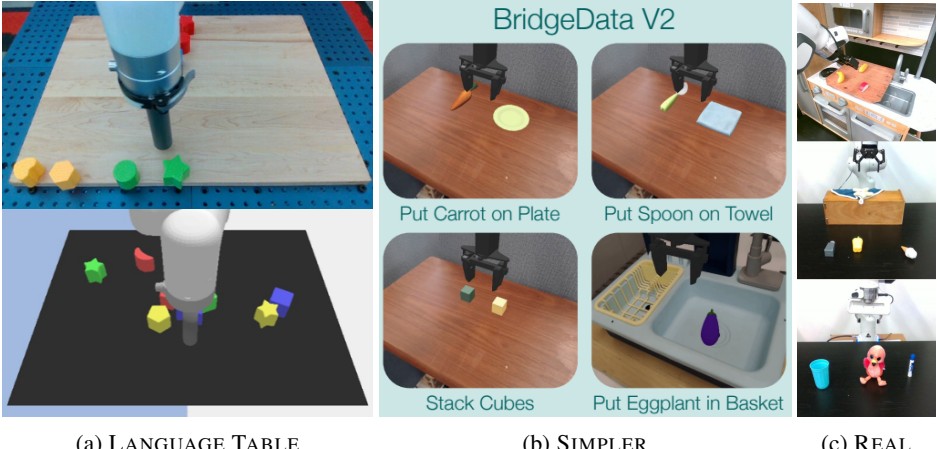

(a) LANGUAGE TABLE         (b) SIMPLER         (c) REAL

Figure 9: **Experimental Setups**. (a) shows an example from the 440k real-world trajectories (top) and the 181k simulation trajectories (bottom) from the Language Table Benchmark. (b) shows the 4 different evaluation tasks we use with the SIMPLER environment. (c) shows the three different tasks that we perform in the real-world.

**Language Table Experimental Setup**   Figure 9 (a) shows examples of the Language Table setup. For Language Table experiments, we train VLA-based models to generate language directions (e.g. 'move up') before actual actions following Belkhale et al. (2024), which significantly improved the performance [6]. For evaluation, we evaluate on 50 evaluation rollouts for each subtask category where the initial locations of the objects are randomized for each evaluation. Further details can be found in https://github.com/google-research/language-table.

**SIMPLER Experimental Setup**   Figure 9 (B) shows examples of the SIMPLER setup. The SIMPLER environment does not provide any fine-tuning data for their evaluation pipeline, Thus, we first train our underlying VLM on the Bridgev2 dataset and perform zero-shot rollout on the 4 tasks in SIMPLER. Note that we use held-out trajectories differing in object orientation and position from the evaluation setup. We filter 25 successful trajectories for each task (total of 100) and use them as the fine-tuning dataset for all of our experiments. For evaluation, we evaluate on 24 rollouts

---

[6]For 7 DOF robot experiments, we found the benefit of generating language directions to be marginal compared to the increased inference cost. Therefore, we only generate delta end-effector actions on other experiments.

per task while randomizing the initial object locations. We consider Bridgev2 and SIMPLER to be *in-domain* since they show a high correlation between real-world and simulation results with their simulation benchmark. Further details can be found in https://github.com/simpler-env/SimplerEnv.

**Real-world Tabletop Manipulation Experimental Setup**   Figure 9 (C) shows examples of the real-world tabletop manipulation experimental setup. For the teleoperation, we use the polymetis robotic stack[7] to collect 150 trajectories for each of the tasks. All of the tasks require multi-instruction following capabilities since there are 3 objects in the scene and the model has to condition on the task description to infer which object to interact with. Figure 10 shows samples of each task. For each task, we aim to quantify 3 distinct capabilities:

(1) We test the ability to infer the correct object from the task description between an unseen combination of seen objects during fine-tuninig, (2) We test the ability to infer the correct object from totally unseen objects during fine-tuning that may or may have not been observed during pretraining. Specifically, the *knocking* tasks was conducted with real-world objects that were highly unlikely to have been in any of the pertaining datasets. (3) We test the ability to infer the correct object (among seen objects, unseen combinations) from a totally unseen instruction that requires semantic reasoning (e.g. Pick up a spicy object). For each evaluation criteria, 6 rollouts are performed for each models, resulting in a total of 18 rollouts for each task category. Since there are three tasks, each model is evaluated with 54 rollouts in the real-world. We provide the full list of all of the seen and unseen objects used for each rollout in Table 13, 14, 15, and the total average success rates in Table 16.

Furthermore, for a fair comparison, we match the image resolution during training of all of our models and use the exact same object initial positions for all of our evaluation, mostly on the same day to minimize variability. For evaluation metrics, we adapt a partial success criteria for fine-grained evaluation, following Kim et al. (2024), which we describe in detail below.

*Knock down the <object>.*

For knocking, we give 0.5 partial score if the robot reaches to the correct object and 1 if the robot knocks down the correct object.

*Cover the <object> with a towel.*

For covering, we give 0.33 partial score if the robot picks up the towel correctly, 0.66 if the robot reaches to the correct object or if the towel partially covers the object, and 1 if the correct object is completely covered by the towel.

*Pick up the <object> and put it in the sink.*

For pick and place, we give 0.25 for reaching to the correct object, 0.5 for grasping the object, 0.75 for grasping and moving the object towards the sink, but failing to place the object in the sink, and 1 for placing the correct object in the sink.

## C   BASELINE DETAILS

For UNIPI, we use diffusion model from (Ko et al., 2024) which can be trained on 4 A100 GPUs. For all experiments, we train with 128 batch. We use the same inverse dynamics model as VPT during inference. To mediate estimation errors between the predicted video plans and executed actions being accumulated, we periodically conduct replanning by regenerating new video plans after executing two actions. For VPT, we use ResNet18 followed by an MLP layer for the inverse dynamics model(IDM). The IDM is trained to predict an action when given two frames on a single A6000 GPU using using Adam optimizer with a learning rate 1e-4. For OpenVLA (Bridge), we pretrain on Bridgev2 for 30 epochs with a batch size of 1024. For OpenVLA (Open-X), we use the pretrained checkpoint from Kim et al. (2024). For finetuning, we use LoRA finetuning (Hu et al., 2022) with batch size of 32. We have observed that full-finetuning and lora finetuning leads to similar performance, so we use LoRA finetuning as default for efficient fine-tuning. We finetune the model until the training action accuracy reaches 95%. For ACTIONVLA and LAPA, we train with a batch size of 128 and with image augmentation for real-world finetuning.

---

[7]https://github.com/facebookresearch/polymetis

# D  EXPERIMENTAL RESULT ANALYSIS

Table 4: **Pretraining trajectories statistics for downstream tasks.** Number of trajectories that are the same task with evaluation task for each pretraining dataset: Bridgev2, Open-X, and Something Something V2 (Sthv2) dataset.

| Task | Bridgev2 | Open-X | Sthv2 |
|---|---|---|---|
| Knocking | 2 | 7,969 | 6,655 |
| Covering | 898 | 5,026 | 6,824 |
| Pick & Place | 10,892 | 911,166 | 3,272 |

We further analyze the real-world robot results shown in Figures 3 and 4b, focusing on how the task distribution in pretraining data impacts downstream performance. Table 4 presents the number of trajectories corresponding to each evaluation task (Knocking, Covering, and Pick & Place) across pretraining datasets (Bridgev2, Open-X, and Something Something V2 (Sthv2)), determined through *lexical* matching. We expect future work to use other methods of analyzing the relationship between pertaining and fine-tuning task distributions that capture *semantics* of the task rather than simple lexical matching. We perform this analysis to get a sense of how the task distribution in the pretraining data affects downstream task performance.

**Knocking**  There are almost no knocking-related trajectories in Bridgev2. This scarcity may explain why models trained on Bridgev2 performed worse compared to those trained on Sthv2, despite a larger embodiment gap in the Sthv2 dataset (Figure 4b).

**Covering**  A similar trend is observed for the covering task. Given that the number of covering trajectories in Bridgev2 is relatively small compared to the Sthv2 dataset, models trained on Bridgev2 occasionally underperform compared to LAPA trained on Sthv2.

**Pick & Place**  For the pick and place task, the trend reverses. The number of pick and place tasks in Sthv2 is relatively small compared to Bridgev2 and Open-X, which might explain why LAPA trained on Sthv2 significantly underperforms models trained on Bridgev2 or Open-X. Based on these results, we expect that pretraining on videos encompassing a wide range of skills will lead to a more robust generalist policy compared to training on robot videos with narrower skill sets. We also expect future research to provide a more in-depth analysis of the relationship between task distribution in pretraining data and performance on downstream tasks.

We also present the win rate of LAPA (Open-X) against OpenVLA (Open-X). As illustrated in Figure 11, LAPA outperforms OpenVLA in 65.4% when disregarding the ties. When considering the ties, LAPA outperforms OpenVLA in 31.5% of cases, while OpenVLA prevails in only 16.7%. Interestingly, they tie in 51.9% of the trials, suggesting that in about half the instances, both models either fail or achieve a similar partial success score. Note that these evaluations were performed while ensuring that the target and distractor objects were in identical initial locations during evaluation, alternating the models during evaluation. These results provide insight into the statistical significance of the comparison, supporting the use of multiple metrics to ensure a more comprehensive evaluation of physical robot performance in real-world scenarios (Kress-Gazit et al., 2024), not only the average success-rate across all of the evaluation rollouts.

# E  DETAILED LATENT ACTION ANALYSIS

We provide further qualitative analysis of LAPA. First, we analyze latent actions learned from Language Table with vocabulary size of 8 and sequence length of 1. In Figure 12, we show that each latent action corresponds to a semantic action (0: Move left and forward, 1: Move left and back, 2: Move right and back, 3: Move right slightly, 4: Move right, 5: Move back, 6: Do not move, 7: Move forward). We observe that increasing the latent action vocabulary size leads to capturing a more fine-grained information. We analyze the relationship between latent actions with ground-truth 2 DOF actions by mapping each instance into latent action space. As shown in Figure 13,

we observe that latent actions are well-clustered in the actual 2D action space, indicating that latent actions are meaningful representations that are highly related to actual continuous actions.

We further analyze the latent actions learned from human manipulation videos using the Something-Something V2 dataset. As illustrated in Figure 14, these latent actions capture not only hand movements but also camera movements. Since the camera viewpoint varies throughout the videos in the Something-Something V2 dataset due to the videos being egocentric, our latent action quantization model also learns to represent camera movements. For instance, latent actions [3,5,2,7] and [5,6,7,6] correspond to slight downward camera movement, [4,0,0,4] and [2,3,6,6] indicate rightward movement, and [4,2,0,0] and [5,7,0,5] represent subtle upward camera shifts.

## F    ADDITIONAL ABLATION RESULTS

We first analyze the effect of window size $H$ for latent action quantization process. For all robot manipulation videos, we have determined the window size depending on the fps of the video so that the next frame models 0.6 seconds ahead from the current frame. For human manipulation videos, we have set the next frame to be 2.4 seconds ahead since we qualitatively observed that many of the frames of the human videos contain much less dynamic actions compared to robot videos. (However, we think that filtering these frames could make the window size the same as robot videos, which we leave as future work.). We have added an ablation experiment on the window size for robot videos (Bridgev2) by evaluating on SIMPLER in Figure 15a. Note that the default is $H = 3$ because Bridgev2 is collected with 5hz. The results show that LAPA is quite robust to different window sizes. However, if the window size is extremely large, performance degradation is observed. This is expected since our quantization model is relatively small (300M parameters), it faces difficulties modeling latent information when the visual deltas are significant.

We also analyze the data scaling in terms of fine-tuning data by comparing with SCRATCH. We evaluate on SIMPLER. As shown in Figure 15b, LAPA (Bridge) consistently outperforms SCRATCH even when the fine-tuning data instances are small while the absolute performance increases with larger fine-tuning data.

For data scaling, we also analyze the data scaling of human videos. We compare LAPA trained from 10% of Sthv2 human video dataset with LAPA trained from the whole Sthv2 human video dataset. Results in Table 12 show that scaling the human video datasets boosts the performance for SIMPLER benchmark not only for the final success for all subtasks. We leave exploring scaling law for human videos more extensively or future work, since showing scaling law requires intensive computational resources to do different ablations of model size, data size, and computational resources.

Finally, we vary the latent action length and vocabulary size in Language Table, extending the result of Figure 5 which was analyzed in Bridgev2 data. As shown in Figure 16, increasing the sequence and vocab size increases the performance. However, unlike SIMPLER, we observe that the increasing the latent action vocab size is much more effective compared to increasing the latent action sequence length in terms of absolute performance. This implies that for environments that are visually simple, increasing the latent action vocabulary might be more effective compared to sequence length.

## G    DETAILED EXPERIMENTAL RESULTS

### G.1    LANGUAGE TABLE

We provide the detailed results of the experiments performed on the Language Table benchmark in Table 5, 6, 7, 8, 9, 10. For all of the tables in the appendix, we **bold** the best result among the comparisons and underline the second best. Each value denotes the success rate (%). 50 evaluation rollouts are performed for each task category, resulting in 250 total evaluation rollouts per model for each table.

We also show the qualitative result of UNIPI where the diffusion model generates the correct plan for simple and short-horizon tasks (e.g. separate tasks). However, the diffusion model generates the wrong plan corresponding to the instruction when the task requires longer horizon planning (Figure 17).

Table 5: **Language Table In-Domain Seen Results.**

|  | SCRATCH | UNIPI | VPT | LAPA | ACTIONVLA |
|---|---|---|---|---|---|
| Block2Block | 4.0 | 14.0 | 36.0 | 58.0 | **76.0** |
| Block2Absolute | 6.0 | 4.0 | 38.0 | 56.0 | **72.0** |
| Block2BlockRelative | 10.0 | 12.0 | 48.0 | 52.0 | **76.0** |
| Block2Relative | 6.0 | 10.0 | 26.0 | 48.0 | **70.0** |
| Separate | 52.0 | 72.0 | 70.0 | **96.0** | 90.0 |
| **AVG** | 15.6 | 22.4 | 43.6 | 62.0 | **76.8** |

Table 6: **Language Table In-Domain Unseen Results.**

|  | SCRATCH | UNIPI | VPT | LAPA | ACTIONVLA |
|---|---|---|---|---|---|
| Block2Block | 8.0 | 4.0 | 26.0 | 50.0 | **62.0** |
| Block2Absolute | 10.0 | 6.0 | 42.0 | 48.0 | **58.0** |
| Block2BlockRelative | 2.0 | 6.0 | 20.0 | 28.0 | **48.0** |
| Block2Relative | 8.0 | 6.0 | 32.0 | 38.0 | **44.0** |
| Separate | 48.0 | 44.0 | 44.0 | **84.0** | 82.0 |
| **AVG** | 15.2 | 13.2 | 32.8 | 49.6 | **58.8** |

Table 7: **Language Table Cross-Task Seen Results.**

|  | SCRATCH | UNIPI | VPT | LAPA | ACTIONVLA |
|---|---|---|---|---|---|
| Block2Block | 18.0 | 12.0 | 74.0 | 74.0 | **76.0** |
| Block2Absolute | 8.0 | 6.0 | 56.0 | 62.0 | **72.0** |
| Block2BlockRelative | 6.0 | 2.0 | 62.0 | 72.0 | **76.0** |
| Block2Relative | 24.0 | 16.0 | **72.0** | 60.0 | 70.0 |
| Separate | 80.0 | 68.0 | 96.0 | **98.0** | 90.0 |
| **AVG** | 27.2 | 20.8 | 72.0 | 73.2 | **76.8** |

Table 8: **Language Table Cross-Task Unseen Results.**

|  | SCRATCH | UNIPI | VPT | LAPA | ACTIONVLA |
|---|---|---|---|---|---|
| Block2Block | 16.0 | 4.0 | **66.0** | 46.0 | 62.0 |
| Block2Absolute | 10.0 | 10.0 | 56.0 | 52.0 | **58.0** |
| Block2BlockRelative | 8.0 | 10.0 | 46.0 | **48.0** | **48.0** |
| Block2Relative | 12.0 | 4.0 | **52.0** | 38.0 | 44.0 |
| Separate | 66.0 | 52.0 | 84.0 | **90.0** | 82.0 |
| **AVG** | 22.4 | 16.0 | **60.8** | 54.8 | 58.8 |

Table 9: **Language Table Cross-Environment Seen Results.**

|  | SCRATCH | UNIPI | VPT | LAPA | ACTIONVLA |
|---|---|---|---|---|---|
| Block2Block | 4.0 | 4.0 | 16.0 | 26.0 | **66.0** |
| Block2Absolute | 6.0 | 4.0 | 8.0 | 16.0 | **58.0** |
| Block2BlockRelative | 10.0 | 8.0 | 6.0 | 20.0 | **62.0** |
| Block2Relative | 6.0 | 4.0 | 12.0 | 22.0 | **54.0** |
| Separate | 52.0 | 48.0 | 48.0 | **84.0** | **84.0** |
| **AVG** | 15.6 | 13.6 | 18.0 | 33.6 | **64.8** |

## G.2 SIMPLER

We provide results of various models evaluated on SIMPLER environment. Table 11 shows the setting where baseline models are pretrained on Bridgev2 and then finetuned on SIMPLER rollouts (100 videos). The results show detailed results for each task (stack green to yellow block, put carrot on plate, put spoon on otowel, put eggplant in basket) and subtasks (grasping and moving). As shown in Table 11, UNIPI significantly underperforms all other baselines on the SIMPLER Environment. We observe that, although the generated plans from the diffusion models are quite

Table 10: **Language Table Cross-Environment Unseen Results.**

|  | SCRATCH | UNIPI | VPT | LAPA | ACTIONVLA |
|---|---|---|---|---|---|
| Block2Block | 8.0 | 2.0 | 2.0 | 30.0 | **38.0** |
| Block2Absolute | 10.0 | 6.0 | 4.0 | 14.0 | **48.0** |
| Block2BlockRelative | 2.0 | 6.0 | 2.0 | 10.0 | **50.0** |
| Block2Relative | 8.0 | 4.0 | 40.0 | 18.0 | **54.0** |
| Separate | 48.0 | 42.0 | 44.0 | 76.0 | **80.0** |
| **AVG** | 15.2 | 12.0 | 18.4 | 29.6 | **54.0** |

accurate, the IDM lacks the capability to predict 7 DOF continuous actions accurately when given only 100 action-labeled trajectories. Specifically, we observe that UNIPI often fails to grasp the object within the maximum step limit. This implies the effectivness of using VLAs in scenarios with insufficient action-labeled data. Similar to the results of Section 4.3, LAPA outperforms baseline models that pretrain without using ground-truth action labels (UNIPI and VPT) and closes the performance gap with ACTIONVLA, which is pretrained on all of the 60K action-labeled trajectories from the Bridgev2 dataset. This highlights the effectiveness of LAPA, even when the complexity of the action space increases. We also evaluate the performance of OPENVLA fine-tuned on SIMPLER trajecotries for reference. The performance of OPENVLA (36.4) is similar to Scratch. The bad performance of OPENVLA on SIMPLER is a well known issue which is due to OPENVLA not being robust to real-to-sim transfer for SIMPLER.

We also provide detailed results of the setting where baseline models are pretrained on human manipulation videos (Something Something V2 dataset) and then finetuned on SIMPLER rollouts (100 videos) in Table 12. We only compare to UNIPI, VPT, and LAPA since ACTIONVLA could not be trained without ground-truth action labels.

Table 11: **SIMPLER results of Bridgev2 Pretraining.** Success, Grasping, and Moving Rates (%) in SIMPLER environment. We pretrain UNIPI, VPT, and LAPA on Bridgev2 dataset without using ground-truth action labels and ACTIONVLA on Bridgev2 using action labels. We also add the result of OPENVLA fine-tuned on SIMPLER trajectories for reference. The main 4 tasks are: stack green to yellow block, put carrot on plate, put spoon on towel, and put eggplant in basket. Best is **bolded** and second best is underlined.

| Success Rate | SCRATCH | UNIPI | VPT | LAPA | ACTIONVLA | OPENVLA |
|---|---|---|---|---|---|---|
| Stack G2Y | 29.2 | 2.7 | 45.8 | 54.2 | **75.0** | 41.6 |
| Carrot2Plate | 29.2 | 2.7 | 37.5 | 45.8 | **58.0** | 50.0 |
| Spoon2Towel | 50.0 | 0.0 | **70.8** | **70.8** | **70.8** | 37.5 |
| Eggplant2Bask | 29.2 | 0.0 | 50.0 | **58.3** | 50.0 | 16.7 |
| **AVG** | 34.4 | 1.3 | 51.0 | 57.3 | **63.5** | 36.4 |
| **Grasping Rate** | | | | | | |
| Grasp Green Block | 66.6 | 20.8 | 62.5 | 62.5 | **87.5** | 50.0 |
| Grasp Carrot | 45.8 | 33.2 | 54.1 | 58.3 | **75.0** | 66.6 |
| Grasp Spoon | 70.8 | 22.2 | 79.2 | **83.3** | **83.3** | 45.8 |
| Grasp Eggplant | 62.5 | 16.0 | 70.8 | **83.3** | 75.0 | 37.5 |
| **AVG** | 61.4 | 23.1 | 66.7 | 71.9 | **80.2** | 50.0 |
| **Moving Rate** | | | | | | |
| Move Green Block | 58.3 | 29.1 | 58.3 | 66.6 | **91.6** | 70.8 |
| Move Carrot | 45.8 | 48.6 | 66.6 | 70.8 | **91.6** | 75.0 |
| Move Spoon | 70.8 | 34.6 | 79.2 | **83.3** | 79.2 | 75.0 |
| Move Eggplant | 87.5 | 58.0 | 70.8 | 87.5 | **91.6** | 50.0 |
| **AVG** | 65.6 | 42.6 | 68.7 | 77.1 | **88.5** | 67.7 |

## G.3 REAL-WORLD

We also provide the full list of objects and the partial success recorded for each of the evaluation rollout: Knocking (Table 13), Covering (Table 14), and Pick & Place (Table 15). The total average success rate is provided in Table 16).

Table 12: **SIMPLER results of Human Manipulation Video Pretraining.** Success, Grasping, and Moving Rates (%) in SIMPLER environment. We pretrain UNIPI, VPT, and LAPA on Something-Something V2 dataset without using ground-truth action labels. The main 4 tasks are: stack green to yellow block, put carrot on plate, put spoon on towel, and put eggplant in basket. Best is **bolded** and second best is underlined.

| Success Rate | VPT | UNIPI | LAPA | LAPA (10%) |
|---|---|---|---|---|
| StackG2Y | **50.0** | 0.0 | **50.0** | 45.8 |
| Carrot2Plate | 29.1 | 1.3 | **50.0** | 41.6 |
| Spoon2Towel | 37.5 | 1.3 | 50.0 | **66.6** |
| Eggplant2Bask | **66.6** | 0.0 | 58.3 | 45.8 |
| **AVG** | 45.8 | 0.7 | **52.1** | 50.0 |
| **Grasping Rate** | | | | |
| Grasp Green Block | **66.6** | 2.7 | 58.3 | 50.0 |
| Grasp Carrot | 45.8 | 31.7 | **62.5** | 41.6 |
| Grasp Spoon | 70.8 | 21.7 | **75.0** | 70.8 |
| Grasp Eggplant | **91.6** | 6.8 | 70.8 | 62.5 |
| **AVG** | **68.7** | 15.7 | 66.7 | 56.2 |
| **Moving Rate** | | | | |
| Move Green Block | **62.5** | 2.7 | **62.5** | 50.0 |
| Move Carrot | 58.3 | 37.5 | **70.8** | 58.3 |
| Move Spoon | 54.1 | 18.1 | 75.0 | **79.2** |
| Move Eggplant | **91.6** | 50.3 | 83.3 | 62.5 |
| **AVG** | 66.6 | 27.1 | **72.9** | 62.5 |

Table 13: **Knocking Task Results**

| | OpenVLA (OpenX) | LAPA (OpenX) | OpenVLA (Bridge) | LAPA (Bridge) | ActionVLA (Bridge) | Scratch | LAPA (Sthv2) |
|---|---|---|---|---|---|---|---|
| **Seen Objects, Unseen Object Combinations** | | | | | | | |
| flamingo | 0 | 0.5 | 0.5 | 0.5 | 0 | 0 | 0.5 |
| pistachios | 0.5 | 1 | 0.5 | 0 | 1 | 0 | 1 |
| soft scrub | 0 | 0 | 0 | 0 | 0.5 | 0 | 0.5 |
| white cup | 1 | 0 | 0 | 0.5 | 0.5 | 0.5 | 0 |
| mustard | 0 | 1 | 0 | 0 | 0 | 0 | 0 |
| water bottle | 1 | 1 | 0.5 | 0 | 0 | 0.5 | 0 |
| SUM | 2.5 | 3.5 | 1.5 | 1 | 2 | 1 | 2 |
| **Unseen Objects** | | | | | | | |
| pringles | 0.5 | 0.5 | 0.5 | 0 | 0 | 0 | 0 |
| hersey's chocolate syrup | 0 | 0 | 0 | 0 | 0 | 0 | 0 |
| popcorn | 0 | 1 | 1 | 1 | 1 | 0 | 1 |
| skittles | 0 | 0 | 0 | 0 | 0 | 0 | 0 |
| green board marker | 0.5 | 0.5 | 0.5 | 0.5 | 0.5 | 0.5 | 0.5 |
| paper towel | 0 | 0 | 0 | 0 | 0 | 0 | 0 |
| SUM | 1 | 2 | 2 | 1.5 | 1.5 | 0.5 | 1.5 |
| **Seen Objects, Unseen Instructions** | | | | | | | |
| a drink that contains orange | 0 | 0 | 0 | 0 | 0 | 0 | 0 |
| food to eat with milk | 0.5 | 0 | 0 | 0 | 0 | 0 | 0 |
| a object used for cleaning | 0 | 1 | 0 | 0 | 0 | 0 | 0 |
| something to wash dishes | 1 | 1 | 0 | 0.5 | 1 | 0.5 | 0 |
| the nuts | 1 | 1 | 0.5 | 1 | 1 | 0.5 | 1 |
| rectangle object | 1 | 1 | 0.5 | 0.5 | 0.5 | 0 | 1 |
| SUM | 3.5 | 4 | 1 | 2 | 2.5 | 1 | 2 |
| Success Rate (Strict) | 27.78% | **44.44%** | 5.56% | 11.11% | 22.22% | 0.00% | 22.22% |
| Success Rate | 38.89% | **52.78%** | 25.00% | 25.00% | 33.33% | 13.89% | 30.56% |
| Reaching Success Rate | 50.00% | **61.11%** | 44.44% | 38.89% | 44.44% | 27.78% | 38.89% |

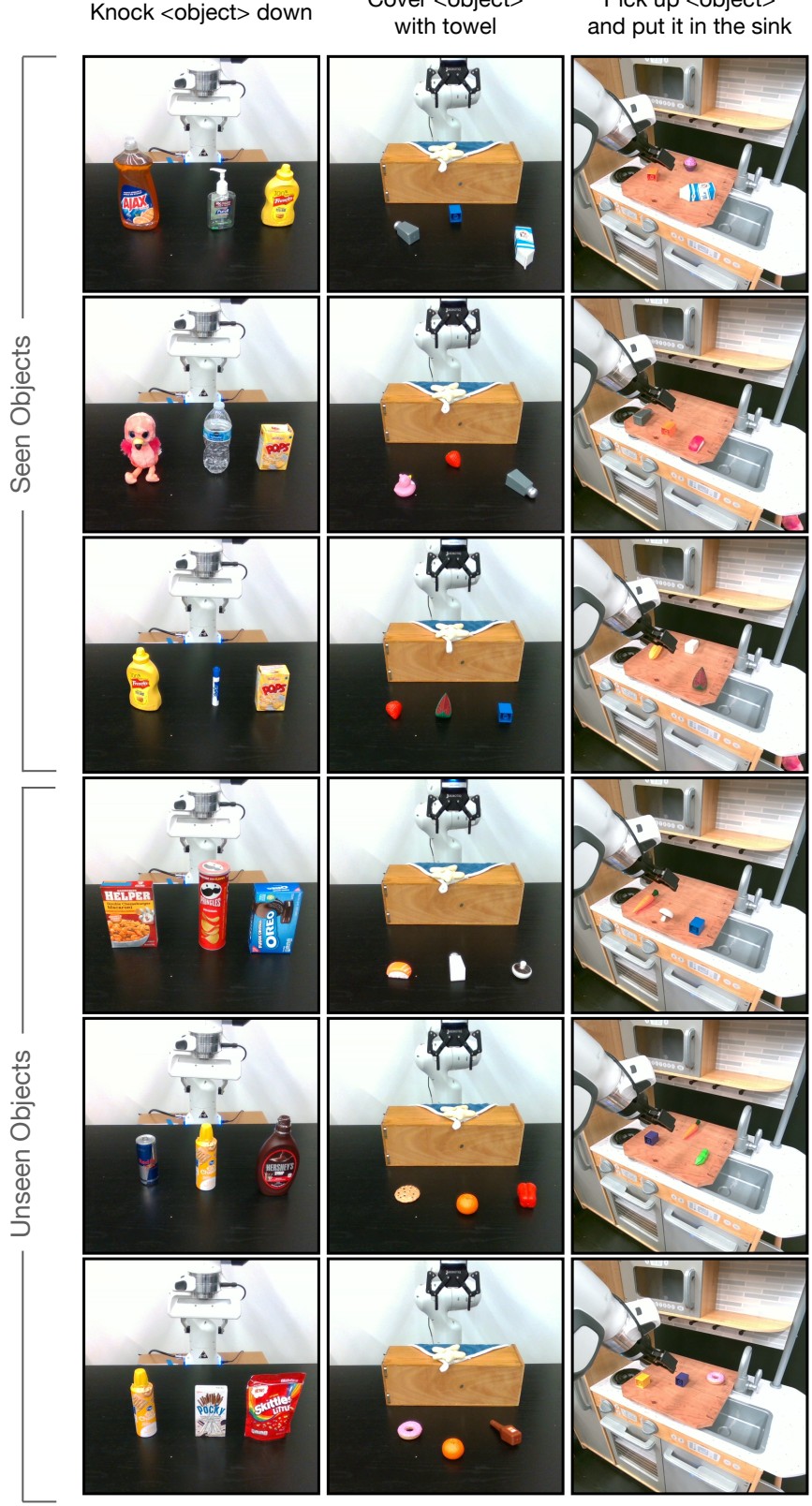

Figure 10: **Real-world Tabletop Manipulation Examples.**

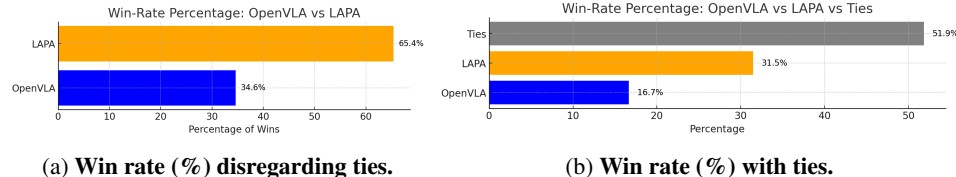

(a) **Win rate (%) disregarding ties.**  (b) **Win rate (%) with ties.**

Figure 11: **Pairwise win rate (%)**. We compare a pairwise win-rate of OpenVLA and LAPA across the 54 evaluation rollouts in the real-world. (a) shows the win-rate while ignoring the ties and (b) shows the ties together with the individual wins.

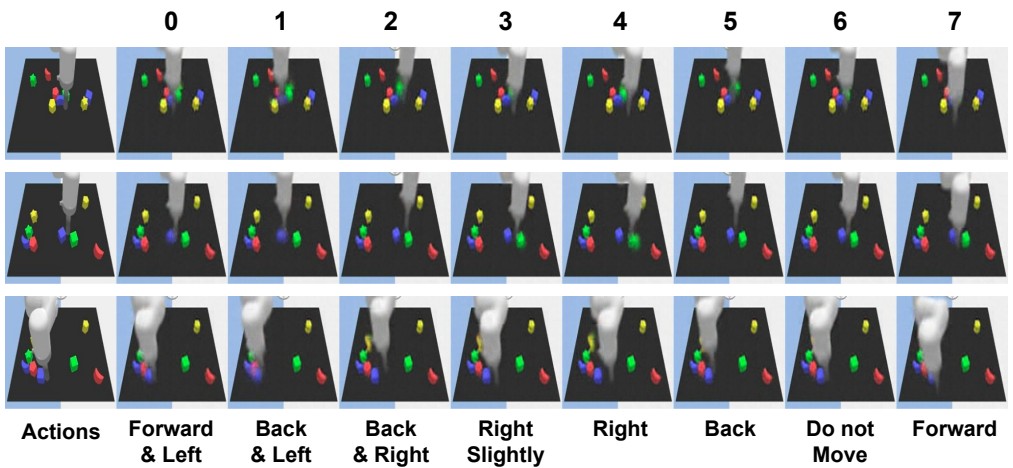

Figure 12: **Latent Action Analysis in Language Table.** We condition the current observation $x_1$ and quantized latent action to the decoder of the latent action quantization model. We observe that each latent action can be mapped into a semantic action. For example, latent action 0 corresponds to moving a bit left and forward and corresponds to moving a bit left and back.

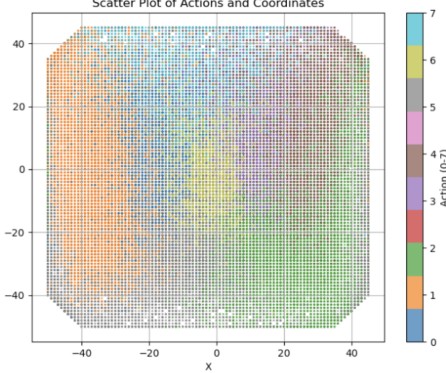

Figure 13: **Correlation of latent action with ground-truth actions** When we map latent actions to ground-truth 2 DOF actions of Language Table, we observe that latent actions are well-clustered in the actual 2D action space.

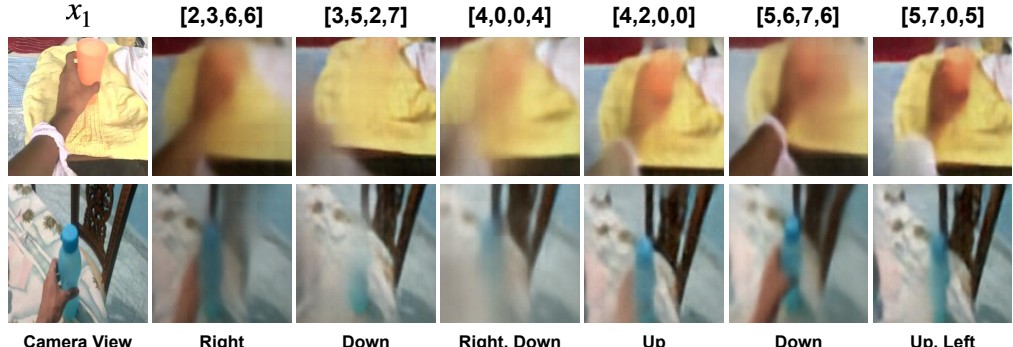

Figure 14: **Latent Action Analysis in Human Manipulation Videos.** We condition the current observation $x_1$ and quantized latent action to the decoder of the latent action quantization model. We observe that each latent action can be mapped into a semantic action including camera movements. For example, latent action [3,5,2,7] corresponds to moving the camera a bit down while [4,2,0,0] corresponds to moving the camera slightly up.

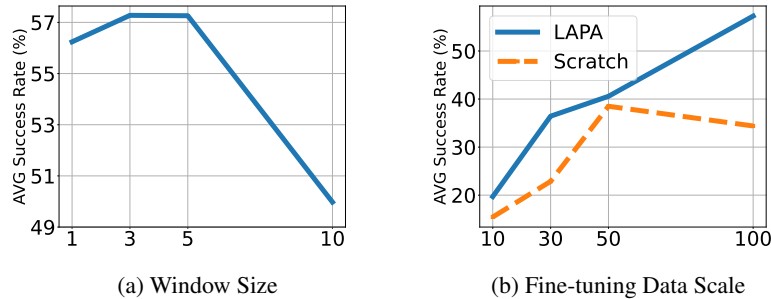

(a) Window Size  (b) Fine-tuning Data Scale

Figure 15: **Additional Ablation Results of LAPA.** We further analyze the performance of LAPA by varying the window size for latent action quantization and fine-tuning data scale. We report the average success rate in SIMPLER.

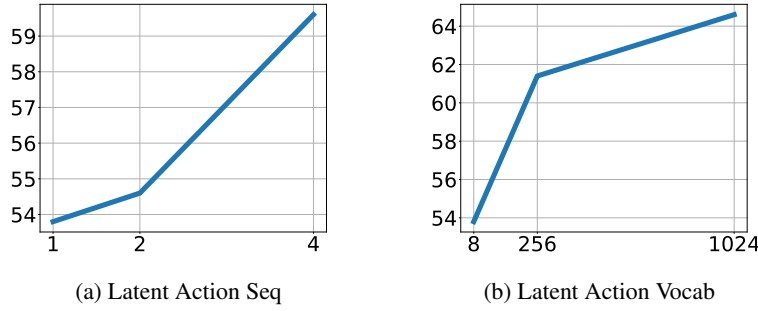

(a) Latent Action Seq  (b) Latent Action Vocab

Figure 16: **Ablation Results of LAPA in Language Table**. We try various latent action vocab and sequences of LAPA and show the downstream average success rate (%) on the Language Table fine-tuning tasks.

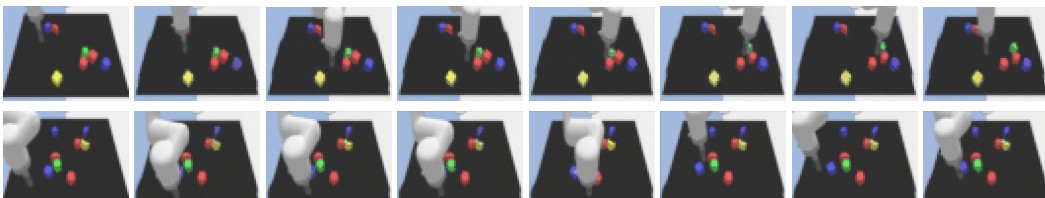

Figure 17: **Success and Failure Cases of UNIPI.** (Top) Given the instruction of 'move the green block away from the red cube and red pentagon', the diffusion model of UNIPI successfully generates the plan. (Bottom) Given the instruction of 'put the blue moon toward the yellow block', the diffusion model fails to generate the correct plan.

Table 14: **Covering Task Results**

| | OpenVLA (OpenX) | LAPA (OpenX) | OpenVLA (Bridge) | LAPA (Bridge) | ActionVLA (Bridge) | Scratch | LAPA (Sthv2) |
|---|---|---|---|---|---|---|---|
| **Seen Objects, Unseen Object Combinations** | | | | | | | |
| icecream | 0.33 | 0.33 | 0.33 | 0.33 | 0.33 | 0.33 | 0 |
| strawberry | 0.33 | 1 | 0.33 | 1 | 0.33 | 1 | 1 |
| pepper | 0.33 | 0 | 0.33 | 0.33 | 0.33 | 0.33 | 0.33 |
| watermelon | 0.33 | 0.33 | 0.33 | 0.33 | 0.33 | 0 | 0.33 |
| blue lego block | 0.66 | 1 | 1 | 1 | 1 | 0.33 | 0.33 |
| pink duck | 0.33 | 1 | 0.33 | 0.33 | 0.33 | 0 | 0.33 |
| SUM | 2.31 | 3.66 | 2.65 | 3.32 | 2.65 | 1.99 | 2.32 |
| **Unseen Objects** | | | | | | | |
| donut | 0.33 | 1 | 0.66 | 1 | 0.66 | 0.66 | 0.33 |
| orange | 0.33 | 0.33 | 1 | 0 | 0.33 | 1 | 1 |
| mushroom | 0.33 | 0.33 | 0.33 | 0.33 | 0.33 | 0.33 | 0.33 |
| yellow lego block | 0.33 | 1 | 1 | 0.33 | 0 | 0.33 | 0.33 |
| peas | 1 | 0 | 0.66 | 1 | 1 | 0.33 | 1 |
| egg | 0 | 1 | 0.33 | 0 | 0.66 | 0 | 1 |
| SUM | 2.32 | 3.66 | 3.98 | 2.66 | 2.98 | 2.65 | 3.99 |
| **Seen Objects, Unseen Instructions** | | | | | | | |
| drink | 0.33 | 0 | 0.66 | 1 | 0.33 | 0.33 | 0.66 |
| yellow object | 0.66 | 0.66 | 0 | 0 | 0.33 | 0 | 0.33 |
| fruit | 0.33 | 0.33 | 0.33 | 0.33 | 0.33 | 0.33 | 0.33 |
| vegetable | 0.33 | 0.33 | 0 | 0.33 | 0.33 | 0.33 | 0.33 |
| edible object | 0.33 | 0.33 | 0.66 | 0 | 0.33 | 1 | 0.33 |
| condiment | 0.33 | 0.33 | 0.33 | 0 | 0.33 | 0.33 | 0.33 |
| SUM | 2.31 | 1.98 | 1.98 | 1.66 | 1.98 | 2.32 | 2.31 |
| Success Rate (Strict) | 5.56% | **33.33%** | 16.67% | 27.78% | 11.11% | 16.67% | 22.22% |
| Success Rate | 38.56% | **51.67%** | 47.83% | 42.44% | 42.28% | 38.67% | 47.89% |
| Reaching Success Rate | 16.66% | **38.89%** | **38.89%** | 27.78% | 22.22% | 22.22% | 27.78% |

Table 15: **Pick & Place Sink Task Results**

| | OpenVLA (OpenX) | LAPA (OpenX) | OpenVLA (Bridge) | LAPA (Bridge) | ActionVLA (Bridge) | Scratch | LAPA (Sthv2) |
|---|---|---|---|---|---|---|---|
| **Seen Objects, Unseen Object Combinations** | | | | | | | |
| milk | 1 | 1 | 1 | 1 | 1 | 0 | 1 |
| orange lego block | 1 | 1 | 0 | 1 | 0 | 0 | 0 |
| ketchup | 0.25 | 0.25 | 0.25 | 0.25 | 0 | 0 | 0 |
| corn | 1 | 0.75 | 1 | 0.25 | 0.25 | 0.25 | 0.25 |
| icecream | 0.25 | 0 | 0 | 0 | 1 | 0 | 1 |
| salt | 0 | 0.25 | 0 | 1 | 0 | 0 | 0 |
| SUM | 3.5 | 3.25 | 2.25 | 3.5 | 2.25 | 0.25 | 2.25 |
| **Unseen Objects** | | | | | | | |
| carrot | 1 | 0.25 | 0 | 0.25 | 1 | 0.25 | 0.25 |
| yellow paprika | 1 | 1 | 0 | 0.25 | 0.25 | 0 | 1 |
| yellow cube | 1 | 0.5 | 0.25 | 0.5 | 0 | 0 | 0 |
| salmon sushi | 0 | 0.25 | 0 | 0.5 | 0 | 0 | 0 |
| orange | 1 | 0 | 0 | 0 | 0 | 0.25 | 0 |
| blue cube | 0.25 | 0.25 | 0 | 0 | 0 | 0 | 0 |
| SUM | 4.25 | 2.25 | 0.25 | 1.5 | 1.25 | 0.5 | 1.25 |
| **Seen Objects, Unseen Instructions** | | | | | | | |
| an object that is yellow | 1 | 1 | 0 | 1 | 0.25 | 0 | 0 |
| an object that is round | 0 | 0.25 | 0 | 0 | 0 | 0.25 | 0 |
| an object that is a fruit | 1 | 1 | 1 | 1 | 0 | 1 | 0.75 |
| an object that you can drink | 0 | 0.25 | 0 | 0.5 | 0 | 0 | 0 |
| an object that is a vegetable | 0 | 0 | 0 | 0 | 0 | 0 | 0 |
| an object that is an animal | 0 | 0.25 | 0 | 0.25 | 0.25 | 0 | 0 |
| SUM | 2 | 2.75 | 1 | 2.75 | 0.5 | 1.25 | 0.75 |
| Success Rate (Strict) | **50.00%** | 27.78% | 16.67% | 27.78% | 16.67% | 5.56% | 16.67% |
| Success Rate | **54.17%** | 45.83% | 19.44% | 43.06% | 22.22% | 11.11% | 23.61% |
| Reaching Success Rate | 66.67% | **83.33%** | 27.78% | 72.22% | 38.89% | 27.78% | 33.33% |

Table 16: **Summary of Total Success Rates (%)**

| | OpenVLA (OpenX) | LAPA (OpenX) | OpenVLA (Bridge) | LAPA (Bridge) | ActionVLA (Bridge) | Scratch | LAPA (Sthv2) |
|---|---|---|---|---|---|---|---|
| Total Success Rate | 43.87% | **50.09%** | 30.76% | 36.83% | 32.61% | 21.22% | 34.02% |
| Total Success Rate (Strict) | 27.78% | **35.19%** | 12.96% | 22.22% | 16.67% | 7.41% | 20.37% |

