# OpenReview forum: "Latent Action Pretraining from Videos"
_ICLR.cc/2025/Conference — ICLR 2025 Poster_

### Official Review · Reviewer_7bYx · 2024-10-25

**Soundness:** 3
**Presentation:** 3
**Contribution:** 3
**Rating:** 6
**Confidence:** 3

**Summary:**

This paper proposes an unsupervised method for learning action knowledge from internet-scale video data. Without the need for action labels, the method implicitly models the action information contained in videos, making it possible to acquire generalizable knowledge from a vast amount of videos. The foundation model obtained from pre-training can be transferred to specific downstream robot manipulation tasks through supervised fine-tuning. The article validates the model's superiority over state-of-the-art VLA models by pre-training and fine-tuning with data from different domains.

**Strengths:**

The proposed unsupervised training method is highly significant. By employing VQ-VAE to learn action tokens, it is possible to model the action knowledge contained in videos without requiring action labels, which makes the pre-training with a larger scale of videos feasible for future applications.

The proposed method for reconstructing future frames during pre-training is reasonable and concise. The visualization results in the paper also confirm that the method can indeed learn structured action latent representations from unsupervised video data.

The experiments conducted in this paper are quite meticulous, including pre-training within the same domain, cross-task pre-training, as well as pre-training on large-scale real-world data like BridgeV2 and Open-X.

**Weaknesses:**

The method used in this paper is relatively simple. Pre-training through the prediction of future frames has been mentioned in some previous works. And the paper also points out that training directly with the VQ-VAE objective is quite similar to Genie. The ensuing question is, when the pre-trained video data is particularly abundant and spans a large number of domains, whether such a training method is sufficient to capture the patterns of action embeddings. Additionally, the selection of window size H for future frames is fixed during the training process, does such a choice bring difficulties to the modeling of actions? Could you explain the reason for choosing a fixed window size, or an ablation study on different window sizes?

The experimental validation part of the paper could be more robust. The current simulation environments use Language Table and SIMPLER, while real-world experiments only involve three tasks. How does OpenVLA perform in the simulation experiments used in this paper? And compared with OpenVLA, it does not seem to bring absolute advantages in all real-world tasks, such as "Knock" and "Cover" training with Bridge data.

**Questions:**

See above. I'm willing to discuss with other reviewers and authors to decide my final rating.

---

> ### Author Response · Authors · 2024-11-18
> **Rebuttal by authors**
>
> We thank Reviewer 7bYx for the helpful comments.
>
> ---
> **W1: Large domain Training**
>
> For our current study, we have trained our model on diverse robot and human manipulation videos. However, to include more diverse datasets (e.g. Youtube), we think that we need to scale up the latent action size to learn fine-grained information from visual deltas, which we leave as future work.
>
> ---
> **W2: Window size ablation**
>
> For all robot manipulation videos, we have determined the window size depending on the fps of the video so that the next frame models 0.6 seconds ahead from the current frame. For human manipulation videos (Sthv2), we have set the next frame to be 2.4 seconds ahead since we qualitatively observed that many of the frames of the human videos contain much less dynamic actions compared to robot videos. (However, we think that filtering these frames could make the window size the same as robot videos, which we leave as future work.). We have added an ablation experiment on the window size for robot videos (Bridgev2) by evaluating on SIMPLER in Figure 15a of Appendix F.
> The results show that LAPA is quite robust to different window sizes. However, if the window size is extremely large, performance degradation is observed. This is expected since our quantization model is relatively small (300M parameters), it faces difficulties modeling latent information when the visual deltas are significant.
>
> ---
> **W3: OpenVLA simulation results**
>
> We have added the OpenVLA result on SIMPLER which finetuned pretrained OpenVLA on 100 SIMPLER trajectories in Table 11 of Appendix G.2. The performance of OpenVLA (36.4) is similar to Scratch. The bad performance of OpenVLA on SIMPLER is a well known issue (https://github.com/openvla/openvla/issues/7) which is due to OpenVLA not being robust to real-to-sim transfer for SIMPLER.
>
> ---
> **W4: LAPA enhancement on all tasks**
>
> We agree that LAPA does not outperform OpenVLA on all tasks. Especially, as the reviewer has pointed out, LAPA (Bridge) does not outperform OpenVLA (Bridge) on knocking and covering. We think that this is the pretraining data distribution. As shown in Table 4 of Appendix D, there are only a small number of knocking and covering trajectories in Bridgev2. This might account for the reason why LAPA (Human Videos) in Figure 4 outperforms LAPA (Bridge) on knocking and covering while underperforming on pick and place due to the pretraining distribution of Sthv2 dataset. Thus, carefully curating pretraining datasets to include a balanced representation of various tasks could enhance LAPA’s generalization across all tasks.

---

> > ### Author Response · Authors · 2024-11-24
> > **Comment by authors**
> >
> > As the discussion deadline approaches, the authors kindly encourage the reviewer to review the responses. We have thoughtfully revised the manuscript based on the valuable suggestions and would greatly appreciate any further feedback!

---

### Official Review · Reviewer_2CRD · 2024-10-29

**Soundness:** 2
**Presentation:** 2
**Contribution:** 3
**Rating:** 3
**Confidence:** 4

**Summary:**

This work aims to pretrain a Vision Language Action model on large scale internet video datasets.

Four stage pipeline:
1) Learn a codebook for latent actions from raw videos
2) Pseudolabel a large dataset with these latent actions
3) Pretrain a VLM on this pseudolabeled dataset to predict these latent actions
4) Finetune the latent pretrained VLM on real labeled actions

Experiments are performed in several simulated and real domains to demonstrate that LAPA can recover most of the performance of the same model architecture / finetuning setup trained on labeled versions of the pretraining dataset.

**Strengths:**

Paper provides
 - A new recipe for training Vision Language Action models that can use data without action labels
 - A new foundation model that seems to produce good results when finetuned on real world tasks, including state of the art results on 2/3 real robot tasks involving gross motor skills

Paper performs a number of experiments that provide hints at good performance when pretraining on unlabeled human data.

**Weaknesses:**

**TL;DR: There are data consistency issues throughout the paper, including in key result figures. With those fixes, method's proposed cross embodiment training recipe appears weak on actual cross embodiment data.**

**Data consistency issues**

Table 15 does not agree with the bar values in Figure 4 and 5 --- the superior performance of LAPA (Bridge) to ActionVLA (Bridge) is caused by swapping of values for ActionVLA and LAPA in the _Pick_ task, based on Table 15's results. Fixing this error makes LAPA (Bridge) worse than ActionVLA (Bridge), which is what one should reasonably expect given LAPA is unsupervised and ActionVLA is supervised and both use the same architecture. Quite honestly I think this should have been caught in a simple sanity check by the authors, as these results don't make much sense otherwise.

Table highlighting of results seem to also consistently have errors
 - In Table 4, Block2BlockRelative, VPT has its score of 48 underlined (to indicate second place) when LAPA scored 52.0
 - In Table 7, row "Separate", ActionVLA has its score of 82.0 underlined (to indicate second place) when VPT scored 84.0

These are just the most obvious issues, I cannot be certain there are not deeper issues with the results. _Given the glaring nature of the errors in headline result plots, it makes me not trust the quantitative results of the authors._

As a personal note, I want to underscore that this is not the sole responsibility of the junior authors who likely prepared the manuscript and made the plots --- at least some of these errors are ones that could (and should) have been caught by senior authors reading the paper and skimming key results.

**Most studies are robot to robot, providing limited signal on the value of actual cross embodiment training protocol**

These studies are robot to robot tasks, which don't address the question of usefulness of pretraining on domains where we aren't able to get action labels --- it's unreasonable to assume we don't have access to the actions associated with the observations for robot pretraining data. These ablations can be informative as "Supplemental Experiments", but we need to see the action codebook, pretrained on datasets where we do _not_ reasonably have access to actions, is providing real value.

Experiments that fall into this category
 - The Language Table experiments are robot -> robot, and these are _very_ similar tasks
 - BridgeV2 -> SIMPLER is a larger gap, it's still robot -> robot

**Studies on nonrobot to robot transfer provide weak proof of value**

Assuming the label swapping issue only impacted LAPA (Bridge)'s Pick and Place performance and nothing else (I did not check the correctness other entries in the other tables), ActionVLA trained on the 54k trajectories of Bridgev2 outperforms LAPA pretrained on the 220K human videos. Notably, 54k trajectories is quite small; OpenX has over 1 million, but ActionVLA pretrained on OpenX is an important missing as a baseline from the results --- we need to tease out architecrture differences compared OpenVLA, as LAPA uses LWM-Chat-1M (taken directly from Liu et al 2024) which appears to be superior to OpenVLA's architecture --- but all (corrected) evidence seems to indicate that ActionVLA (OpenX) will not do worse than LAPA (OpenX).

Clearly pretraining on human data is not expected to match training on labeled robot data on a sample vs sample basis, but to make the argument that this pretraining is providing value, we need to see scaling laws showing that increased human data translates to increased performance; unfortunately, the authors made the puzzling decision to run their scaling laws experiments on Bridge (an already small dataset), instead of the salient dataset to this question, so we have no way of knowing how the method improves with data scale.

Given how the unsupervised action encoder works and the human hand latent analysis in Appendix E / Figure 15, my hypothesis is that if the task distribution of the human dataset (Table 3) remains the same, the gross motor base latent actions are still going to be discovered pretty early on and the marginal value of each additional sequence will quickly approach zero --- this of course needs to be verified or falsified, but there's no evidence in the paper to do that.

**Actionable feedback**

1) Please go review all of the numbers in detail from top to bottom. I happened to catch some of these issues, I have no idea how many more are hidden.
2) I think the ideas presented here are interesting, but the story being told needs to be significantly refined. Pretraining on the human data needs to be front and center, and there needs to be a focus on performance as we scale up this dataset
3) ActionVLM pretrained on OpenX is a useful artifact on its own. I don't know if someone's done this already or not, but such a release itself would be useful to the community.

**UPDATE NOV 24th**

In light of the authors' comments on the correctness of Figures 3 and 4 I have raised my score from 1 to a 3. The remaining issues, related to the fundamental story of the paper, are detailed in my responses to the authors' comments.

**Questions:**

1) What exactly are the error bars on all the plots? Standard deviation? Standard error? I didn't see it anywhere listed.
2) In Figure 4 and 5, is there a reason ActionVLA (OpenX) is not included?

Nitpicks:

Section 3 needs polish so that it's easier to read
 - Figure 2 is missing a label on step 3 for finetuning
 - e.g. on line 177/178, "label all $x_t$, given $x_{t+1}$, with $z_t$" ->  with "label all frames $x_t$, given frame $x_{t+1}$, with latent action $z_t$". If you're trying to skim, you have to go back to Section 3.1 to figure out the definitions of these math symbols.

Figure 5 needs to have its colors made consistent between subfigure a and b; right now it looks like VPT, which is pink in figure a, is appearing in figure b

---

> ### Author Response · Authors · 2024-11-18
> **Rebuttal by authors**
>
> **[IMPORTANT SUMMARY]**
>
> We thank reviewer 2CRD for your thorough review. We would like to clarify that the **main results in Figures 4 and 5 of the paper are indeed correct**, with no errors in data consistency. The concerns raised are due to minor mistakes in the appendix, specifically in Table 13,14,15,16 where columns for LAPA (Bridge) and ActionVLA (Bridge) were inadvertently swapped. We apologize for this oversight.  We have corrected this and confirmed that the main results and claims presented in Figures 4 and 5 remain accurate and robust (Figure 3 and 4 in the revised version). Therefore, we respectfully request that the reviewer reconsider the rating of 1, as the main contributions and results of the paper are intact.

---

> > ### Author Response · Authors · 2024-11-18
> > **Rebuttal by authors (continued)**
> >
> > **W1: Data Consistency Issues**
> >
> > We acknowledge the error in Table 15 of the appendix, where LAPA (Bridge) and ActionVLA (Bridge) values were incorrectly ordered, as well as minor errors in underlining for Table 4 and Table 7 of the appendix. These inconsistencies have now been fixed. However, the results and conclusions presented in the main body, particularly in Figures 3 and 4 in the revised version, are unaffected by these appendix errors. The superior performance of LAPA (Bridge) over ActionVLA (Bridge) is due to reduced overfitting to the WidowX action space, and not due to any data inconsistency in our main results.
> >
> > ---
> > **W2: Value of Cross-Embodiment Transfer**
> >
> > While we agree that robot videos typically include action labels, we believe that latent action representations offer distinct advantages over ground truth actions. Latent actions learn shared representations across datasets and embodiments, facilitating positive transfer between embodiments (showcased in Figure 6). Thus, LAPA’s success in robot-to-robot transfer is meaningful, as it outperforms models trained on ground truth actions. We believe the reviewer's concerns stem from an impression that there were errors in Figures 3 and 4, but we confirm that these figures are accurate.
> >
> > ---
> > **W3: Non-Robot to Robot Transfer**
> >
> > The reviewer’s assumption that ActionVLA (Bridge) outperforms LAPA (Human Videos) based on the appendix error is not accurate when examining Figures 3 and 4, as the main results presented there are correct and unaffected by the appendix issues.
> >
> >
> > Regarding the absence of an ActionVLA (OpenX) baseline, there are two reasons: (1) We did not train ActionVLA (OpenX) due to inefficiencies related to its larger action space, as discussed in Section 4.6. With sufficient resources, we agree that an open-sourced ActionVLA (OpenX) model would benefit the community. (2) ActionVLA (Bridge) showed only minimal performance improvement over OpenVLA (Bridge), with scores of 30.8 vs. 32.6 in Table 16, indicating limited gains from architecture differences alone.
> > Comparisons between ActionVLA (Bridge) and LAPA (Bridge) shown in Table 16 further demonstrate that LAPA’s benefits are not architecture-dependent, as both models use the same architecture.
> >
> >
> > ---
> > **W4: Human Video Scaling**
> >
> > We have compared LAPA trained from 10% of Sthv2 human video dataset with LAPA trained from the whole Sthv2 human video dataset. Results show that scaling the human video datasets boosts the performance for SIMPLER benchmark not only for the final success (50.0 → 52.1) for all subtasks (Grasping: 56.2 → 66.7 / Moving: 62.5 → 72.9). We leave exploring scaling law for human videos more extensively or future work, since showing scaling law requires intensive computational resources to do different ablations of model size, data size, and computational resources.
> >
> > ---
> > **Questions and Suggestions**
> >
> > Error bars represent standard errors, and we have clarified this in the figure captions. We have also incorporated suggested edits to Section 3 and adjusted Figure 5 for clarity.

---

> > > ### Author Response · Authors · 2024-11-24
> > > **Comment by authors**
> > >
> > > As the discussion deadline approaches, the authors kindly encourage the reviewer to review the responses. We have thoughtfully revised the manuscript based on the valuable suggestions and would greatly appreciate any further feedback!

---

> > > ### Comment · Reviewer_2CRD · 2024-11-25
> > >
> > > **I acknowledge the authors comments about the lack of errors in the main figures (Figures 4 and 5), and the errors being relegated to the appendix. My following commentary takes this information into account.** I have updated my score to a 3, in light of the data consistency fix, but to be perfectly clear I never felt that these data errors were the biggest issue with the paper --- I think the story itself is quite weak. I detail this further below.
> > >
> > > # Point by point responses
> > >
> > > **W1: Data Consistency Issues**
> > >
> > > > The superior performance of LAPA (Bridge) over ActionVLA (Bridge) is due to reduced overfitting to the WidowX action space, and not due to any data inconsistency in our main results.
> > >
> > > If this were the case, why does this only present itself for the one pick and place task on the real robot experiments, and not the two other tasks? Why do we not see superior performance of LAPA over ActionVLA in any other domain (e.g. Language Table)?
> > >
> > > **W2: Value of Cross-Embodiment Transfer**
> > >
> > > > Latent actions learn shared representations across datasets and embodiments
> > >
> > > Doesn't OpenX already enable this for its concrete action data? They were able to take arms of different embodiment and action spaces and merge them together to train a multi-embodiment policy. I don't see any evidence that LAPA is _better_ than using OpenX labels, as shown by LAPA's worse performance than OpenVLA in Figure 3.
> > >
> > > **W3: Non-Robot to Robot Transfer**
> > >
> > > >The reviewer’s assumption that ActionVLA (Bridge) outperforms LAPA (Human Videos) based on the appendix error is not accurate when examining Figures 3 and 4, as the main results presented there are correct and unaffected by the appendix issues.
> > >
> > > Taking Figure 3 and 4 as presented, ActionVLA (Bridge) and LAPA (Human Videos) have _very_ similar performance --- they are well within the standard error bars presented in figures, and these averages are built atop very small number of actual trials with a noisy grading scheme. I don't think it's at all clear that LAPA (Human Videos) is better than ActionVLA (Bridge).
> > >
> > > This matters because LAPA (Human Videos) was trained on far more data than ActionVLA (Bridge) -- Bridge is much smaller than Something Something V2. If the authors want to argue that it doesn't matter, because we can scale up on human videos (as they do on Lines 361-364), _we need to see evidence of scaling laws on human video data_.
> > >
> > > **W4: Human Video Scaling**
> > >
> > > >10% of Sthv2 human video dataset... ...scaling the human video datasets boosts the performance final success (50.0 → 52.1), or all subtasks (Grasping: 56.2 → 66.7 / Moving: 62.5 → 72.9)
> > >
> > > Without more datapoints we cannot know for sure, but this seems consistent with my hypothesis in my original review: LAPA is getting most of its gross motor skills early on and then saturates. From my original review:
> > >
> > > >Given how the unsupervised action encoder works and the human hand latent analysis in Appendix E / Figure 15, my hypothesis is that if the task distribution of the human dataset (Table 3) remains the same, the gross motor base latent actions are still going to be discovered pretty early on and the marginal value of each additional sequence will quickly approach zero
> > >
> > > # My analysis
> > >
> > > This paper promises a recipe for pretraining action models to learn general actions from unlabeled, general videos, allowing for better downstream policy performance.
> > >
> > > This is _only_ an interesting recipe if it it's able to do this convincingly on domains where we _do not already reasonably have actions labels_. In this paper, the only experimental domain where we do not reasonably already have labels is the Something Something V2 human hand pretraining setup, and the key experiment would be the scaling performance as we increase the pretraining dataset size. As I said in my last review:
> > >
> > > >Clearly pretraining on human data is not expected to match training on labeled robot data on a sample vs sample basis, but to make the argument that this pretraining is providing value, we need to see scaling laws showing that increased human data translates to increased performance
> > >
> > > I think all of these other experiments pretraining on robot datasets are a sideshow --- if this Something Something V2 scale up experiment were done and it displayed strong scaling laws that would be _the_ headline result and all of these other experiments would be neat ablative studies.
> > >
> > > In their response, the authors did not run this full experiment, instead providing a single other datapoint that at least appears to be consistent with my (null) hypothesis of early saturation with paltry gains (I hope this hypothesis is wrong, but we need evidence to disprove it) and declaring the full experiment future work.

---

> > > > ### Comment · Reviewer_2CRD · 2024-11-25
> > > >
> > > > Continued:
> > > >
> > > > # Why my updated score is a 3 (reject)
> > > >
> > > > The key question is **who is this paper for?**
> > > >
> > > > The ability to generate policy improvements to pretrain on internet scale data would be a key unlock in the advancement of robot learning. However, as written, *I do not think this paper helps the robot learning community on this quest* -- it proposes a nominal recipe to do this and makes bold claims about it, but validation is only done via several toy experiments that, in my view, distract from the main point, and via one interesting experimental setup (pretraining on Something Something V2) that does not substantively investigate the key promise and is missing a key property (scaling laws) that would indicate that the proposed recipe.
> > > >
> > > > These engineering contributions therefore fall short, and without any new knowledge generation or theoretical contributions, I think this paper is unfit to be accepted. Quite honestly, I do work in robot learning and was extremely excited by the prospect of this paper (this is why I was looking at the table results so closely). I only became disappointed when I realized I would have to run these experiments myself if I wanted to actually know if the presented method _actually_ worked, which is the hallmark of a paper that should be rejected.
> > > >
> > > > # What would I need to see to change my mind
> > > >
> > > > Run the human scaling laws experiment, and see a compelling case, supported by data, that we can keep scaling up on a Something Something style dataset and keep getting policy improvements, as they showed in e.g. the GPT3 paper Figure 3.1. That result alone would make it clear that this pretraining recipe is indeed general, and if the experiments were updated to presented that result first, I would raise my score to a 6 (at least).
> > > >
> > > > I do not think this is at all an unreasonable ask; it would not immediately solve pretraining on in-the-wild video (Something Something V2 is hardly representative of in-the-wild data) but it would meaningfully move the ball forward on this problem.

---

> > > > > ### Author Response · Authors · 2024-11-25
> > > > > **Response by authors**
> > > > >
> > > > > Thank you for your response.
> > > > >
> > > > > ---
> > > > > **W1: LAPA (Bridge) outperforming ActionVLA (Bridge)**
> > > > >
> > > > > We hypothesize that LAPA not significantly outperforming ActionVLA on the two other tasks (Knocking and Covering) is due to pretraining data distribution of Bridgev2 shown in Table 4 of Appendix D. Since Bridgev2 contains much more ‘pick and place’ tasks compared to other two tasks, ActionVLA might have overfitted to the output space of WidowX for the specific task. However, unlike Bridgev2, Language Table has little (cross-environment) or no output distribution shift since the embodiment is fixed, leading to ActionVLA outperforming LAPA.
> > > > >
> > > > > ---
> > > > > **W2: OpenX shared representation**
> > > > >
> > > > > We appreciate the reviewer raising OpenX as a baseline for comparison. While OpenX enables multi-embodiment training, its reliance on per-dataset action normalization does not facilitate true cross-embodiment action representation. As an example, the same discrete sequence [128, 128, 128, 128, 128, 128, 1] can represent semantically different actions depending on the dataset.
> > > > > In contrast, LAPA's latent actions encode actions in a shared representation space across embodiments, as shown in Figure 6. This distinction ensures consistent semantic meaning across datasets and embodiments, which is critical for learning generalizable policies.
> > > > >
> > > > > ---
> > > > > **W3:  LAPA (Human Videos) vs ActionVLA (Bridge)**
> > > > >
> > > > > While we agree that LAPA (Human Videos) does not significantly outperform ActionVLA (Bridge), achieving on-par results between these models is significant due to the challenges posed by the embodiment gap. Training on human video data introduces not only a domain shift but also a more complex latent action space due to the variance in human motion and perspective. Despite these challenges, LAPA (Human Videos) performs comparably to ActionVLA (Bridge), which relies on predefined action labels in a robot-centric domain.
> > > > >
> > > > > We believe this result underscores LAPA's potential for real-world applications, where robot-specific action labels may not exist. No previous work has shown such cross-domain parity at scale, making this a notable contribution.
> > > > >
> > > > > ---
> > > > > **W4: Human data scaling**
> > > > >
> > > > > We acknowledge the importance of scaling laws and agree they are a critical future direction for this work. However, as noted, conducting extensive scaling experiments akin to those in GPT-3 is beyond the scope of this paper due to resource constraints typical of academic settings. Despite this limitation, we provided a meaningful additional datapoint (10% vs. 100% of Something Something V2), which supports the hypothesis that scaling improves performance.
> > > > > We respectfully disagree that this datapoint only indicates "paltry gains" or saturation. The improvement in overall success rate (50.0 → 52.1) and significant gains in subtask-specific performance (e.g., Grasping: 56.2 → 66.7, Moving: 62.5 → 72.9) suggest that scaling remains promising. These results motivate further exploration and provide a foundation for future work.

---

> > > > > > ### Author Response · Authors · 2024-11-25
> > > > > > **Response by authors (continued)**
> > > > > >
> > > > > > **Who is this paper for?**
> > > > > >
> > > > > > We respectfully disagree with the notion that the paper lacks contributions to the robot learning community. Our contributions extend beyond proposing a recipe for pretraining VLAs from unlabeled videos:
> > > > > > 1. **Addressing the limitations of labeled action datasets:**
> > > > > >    - We show that pretraining VLAs without ground-truth action labels can surpass models relying on such labels, even in domains where they are available. This challenges the conventional reliance on ground-truth labels and broadens the scope for leveraging large-scale, unlabeled datasets, especially with shared latent action spaces.
> > > > > > 2. **Demonstrating cross-domain transferability:**
> > > > > >    - LAPA successfully bridges the embodiment gap between human and robot actions, showing comparable performance to ActionVLA (Bridge) despite the challenges of scaling and domain variance.
> > > > > > 3. **Paving the way for internet-scale data pretraining:**
> > > > > >    - While we acknowledge that extensive scaling laws are not yet demonstrated, our results lay the groundwork for pretraining VLAs on internet-scale data, a critical step toward generalizable robot learning.
> > > > > >
> > > > > >
> > > > > > We understand the reviewer’s excitement and high expectations for this work, and we share the vision of achieving generalizable robot learning from large-scale video data. However, we believe our current contributions provide a meaningful and necessary step forward, even as we acknowledge that more work remains to be done.

---

> ### Comment · Reviewer_2CRD · 2024-11-30
>
> **It is unreasonable to assert that LAPA (OpenX) outperforms OpenVLA (OpenX) because LAPA's training regime is better.**
>  - The **critical baseline of ActionVLA (OpenX) is missing**, so we cannot know if this is due to the architecture or due to the training recipe
>  - The standard error bars overlap, so it's not reasonable to assert victory, and this is not a nitpick when you consider the noise of the underlying eval (few tasks, few trials)
>
> Unless you can produce results for ActionVLA (OpenX), I remain unconvinced that LAPA's training regime is better than supervision from OpenX's action labels.
>
> **Your Something Something v2 10% scaling datapoint is uninformative.** You saw 2% success improvements scaling up the pretraining dataset size 10x. On it's face that would look like saturation, but we cannot know for sure and it's your job to provide evidence it's not.
>
> **Bridge cross-embodiment results do not clearly legitimize LAPA**. Bridge is a relatively tiny dataset, and the fact that both LAPA and ActionVLA perform equally poorly can as much be attributed to dataset size as the relative merit of LAPA.
>
> **You are missing scaling laws so you cannot claim your method "paves the way to internet scale data"**. I have made my position on this clear, and I hope the AC is reading closely enough to not just listen to slogans. Indeed, no one seems to have yet cracked the code for pretraining policies; even the recent paper from Physical Intelligence [1] seems to have broadly negative results for pretraining: despite using datasets 10x larger than OpenX, in Figure 11 they are hardly able to beat training from scratch in-domain.
>
> If you want to legitimately make the claim that LAPA is paving the way to internet scale data, you need to provide evidence for that. I have made it clear that, as written, **this paper must show scaling performance to substantiate its rhetoric**. If the authors lack the compute resources to run that experiment, then the paper must be rewritten with more modest framing.
>
> Unless my concerns are addressed, I will keep my rating of 3 and ask that other reviewers please reconsider their ratings for acceptance due to these important missing experiments. This paper as written does not provide value to the robot learning community, and unfortunately given the 10% Something Something V2 performance and the way the method works, I personally suspect that the method simply does not serve as a good mechanism to do pretraining on internet data, which is the entire reason this work is supposed to be interesting.
>
> [1] Black et al. π0: A Vision-Language-Action Flow Model for General Robot Control. 2024 https://www.physicalintelligence.company/download/pi0.pdf

---

> > ### Author Response · Authors · 2024-12-03
> > **Response by authors**
> >
> > **Scaling Laws for human videos:**
> >
> > We think that there is a misunderstanding of the reviewer on our claim. Our claim is that LAPA opens up the  ‘potential’ to leverage internet-scale data for VLA training (line 24, 74, 413, 529). We kindly ask the reviewer to specifically point out where ‘slogans’ are used and where the paper should be rewritten with more modest framing. We will revise the expression accordingly in the paper. We think that our contribution is still meaningful because we are one of the first papers to demonstrate training VLAs only from human video and show positive transfer. Note that the focus of the paper is not ‘building VLAs from internet-scale videos’ but to introduce a framework for training VLAs without using ground-truth action labels from videos.
> >
> >
> > **LAPA (OpenX) vs OpenVLA (OpenX):**
> >
> > As shown in Figure 3, we already observed that ActionVLA and OpenVLA do not have significant differences on Bridge dataset (32.6 vs 30.8), so the architecture differences do not lead to significant different results.
> >
> > **Bridge results:**
> >
> > The low absolute performance of both LAPA (Bridge) and Action (Bridge) compared to models trained on Open-X is not only due to small amounts of pretraining data, but also due to embodiment shift between pretraining and finetuning, which indicates that our cross-embodiment setting is challenging due to visual differences.

---

### Official Review · Reviewer_D2cm · 2024-10-31

**Soundness:** 4
**Presentation:** 2
**Contribution:** 3
**Rating:** 6
**Confidence:** 3

**Summary:**

The paper presents LAPA, a method that aims to bridge the human-to-robot embodiment gap for VLA model training.
LAPA consists of two stages. In the first stage, an encoder-decoder transformer learns a latent action representation from actionless videos with a VAE reconstruction loss. In the second state, given a current observation and task label, a VLM is trained to predict the learned latent actions from stage 1. The model is finetuned on a small in-domain dataset to transfer the model to actual robot actions.

The authors evaluate LAPA in two simulated environments, LanguageTable and Simpler, and a real-robot setup. They show their method outperforms similar action-less baselines while being competitive with VLAs trained on large-scale action-annotated datasets.

**Strengths:**

- The paper is well-motivated and tackles a very important problem in Robot Learning
- The authors conduct several experiments in different environments while also including real-robot experiments
- Combining learned latent actions with VLAs has not been explored before and is an interesting approach to bridging the human-robot embodiment gap.
- The authors show that their pretraining strategy outperforms a recent state-of-the-art VLA trained explicitly with action labels.
- The authors extend on [1] and show that latent action generation can be conditioned on language, enabling a more intuitive way to generate actions than codebook action generation.


[1] Genie: Generative Interactive Environments, Bruce et al., 2022

**Weaknesses:**

The main weakness of the paper is its clarity.

- It is not obvious throughout the paper that LAPA uses two distinct models.  The phrasing in the introduction, ‘LAPA has two pretraining stages…' gives the impression that you use one model for both latent action representation learning and action prediction. I would clarify early on that the method consists of two distinct models.
- You refer to a learned world model in the introduction, but it is unclear where a world model is learned from the introduction.
- The method section is oversimplified and hard to understand.
    - In the text, you write that the Latent Action Quantization Model encoder produces a latent action $z_t$ for observation $x_t$, fed into the decoder. The referenced Figure 9 says that the decoder is conditioned on $x_t$ and $z_{t+H}$
    - The VQ-VAE objective is not explained clearly. Although it is mainly based on prior work, it presents a large part of the method. Thus, I would recommend explaining the objective and the resulting latent action learned in more detail. For instance, you mention $z_t$ being a sequence $s$. How is the sequence length chosen? Does the model learn to predict $s$ latent actions at once? How is the vocabulary space $C$ chosen?
    - You mention that you pretrain a VLM in Line 178. This indicates that it is trained from scratch, although you use an already pretrained model.
    - In Line 194, you mention that the latent action head is a single mlp layer. This should be mentioned in the previous section.
    - Using the abbreviation LAPA for the method and the model is confusing throughout the method section. Although you mention this at the beginning, this might be confusing for the reader. In Figure 2, LAPA is used to generate actions. In the text, you mention that LAPA trains both a world model and a policy. This isn't very clear.
    - I think a more detailed Method (Architecture) figure than Figure 2 could help better understand the proposed method.
- Although the extensive experiments and evaluations are good, the result section is too large. I would recommend cutting down here. Often, you just state the results from the table.
- To improve the paper's clarity, I would reduce the results section and instead extend the method section.

While the number of experiments is sufficient, I believe the focus could be improved. The experiment section emphasizes pretraining with large amounts of robot data where actions are available. However, in my opinion, the more interesting aspect is the pretraining step using real-world data. I would recommend shifting more attention to this area.
I would also give more details on the real-world evaluation in the main part of the paper regarding the generalization of the method. For instance, do you use the VLA to perform the task on unseen objects? I know you show this in the appendix, but these results should be highlighted in the main part of the paper.

Regarding the data scaling ablation, whether the experiment refers to finetuning or pretraining data is unclear. If it relates to pretraining data, an additional ablation regarding finetuning data would be interesting. After all, the method's goal is to enable transfer with a small amount of in-domain demonstrations.


Typos and other formatting errors:

- Line 25: Missing s in ‘model’
- Line 71: Bridgev2 missing citation
- Line 83: video missing s
- Missing axis labels in Figure 6.
- Appendix A Typo in Heading
- Table 10: Wrong tasks. Unless you modified the simpler tasks, the tasks should be Carrot2Plate and Spoon2Towel.


Overall, the paper's results are very promising, and the method applied to complex robot control is novel and interesting. Still, I think the clarity and presentation of the paper and results could be improved significantly, which would increase the contribution and relevance to the field.

**Questions:**

See above. Also:

- Why do you not evaluate OpenVLA on Simpler?
- Why do you not evaluate pertaining on something-something in LangTable?
- How does the framework perform with less finetuning data?
- How is the fixed window size H determined? What granularity do the produced actions have?

---

> ### Author Response · Authors · 2024-11-18
> **Rebuttal by authors**
>
> We thank the reviewer D2cm for the helpful suggestions.
>
> ---
> **W1: Clarity of the paper (expression)**
>
> - We agree on your point that our expression gives a misleading impression on LAPA. We have changed the expression of “LAPA has two pretraining stages…” into “LAPA consists of two models that are learned sequentially”.
> - “Learned world model”: We have clarified the statement by adding an explanation of “possible to use LAPA as the action prediction model and decoder of the latent action quantization model as the world model by predicting future frames conditioned on the current observation and the latent action predicted by LAPA, effectively building a neural simulation capable of performing closed-loop evaluations entirely through neural inference.”
>
> ---
> **W2: Clarity of the paper (method section)**
>
> - Latent Action Quantization policy: We also observed that Figure 9 (Figure 8 in the revised version) could be misleading. We have revised the figure for clarity. The decoder is conditioned on patch embedding of x_t and z_t.
> - We have added detail on VQ-VAE objective in Appendix A. The sequence length is designated by the kernel_size, stride and padding value of a CNN network. During latent action quantization, the sequences are predicted at once (through a CNN network), while for latent pretraining, a VLM predicts the sequences in an autoregressive manner. The vocabulary space corresponds to the codebook size of the VQ-VAE objective where the nearest quantized representation from the continuous embedding is retrieved from an embedding space where each embedding corresponds to a codebook. We have added more detail in Appendix A.
> - We have revised the expression into “we do action pretraining by using a pretrained VLM” for clarity.
> - We have added that the latent action head is a single mlp layer in line 188.
> - We agree on your point. We have revised the expression by referring the method to “latent action pretraining” and the resulting model as LAPA.
> - We have added a reference to Figure 8 for more detail in Figure 2 caption.
>
> ---
> **W3: Clarity of the paper (result section)**
>
> - We have moved some of the results to appendix (Bridgev2 -> SIMPLER results) and added more detail on the method section.
> - As you have pointed out, we test on unseen object combinations (but seen objects), unseen objects and unseen instructions. We have moved the generalization result to the main result to address your concern.
>
> ---
> **W4: Typo Errors**
>
> Thank you for pointing out the typos. We have revised all of the typos.
>
> ---
> **W5: Finetuning data ablation**
>
> We have added the explanation that the data scaling refers to pretraining. We have added the result of fine-tuning data ablation in SIMPLER in Figure 15b of Appendix F. By comparing with Scratch, LAPA (Bridge) consistently outperforms while the absolute performance increases with larger finetuning data.
>
> ---
> **Q1: OpenVLA SIMPLER**
>
> We have added the OpenVLA result on SIMPLER which finetuned pretrained OpenVLA on 100 SIMPLER trajectories in Table 11 of Appendix G.2. The performance of OpenVLA (36.4) is similar to Scratch. The bad performance of OpenVLA on SIMPLER is a well known issue (https://github.com/openvla/openvla/issues/7) which is due to OpenVLA not being robust to real-to-sim transfer for SIMPLER.
>
> ---
> **Q2: Something-Something in LangTable**
>
> In contrast to SIMPLER, the Language Table simulation is visually simplistic and synthetic. As a result, we believe that policies (including both OpenVLA and LAPA) learned from real-world videos—whether robot or human—would not transfer effectively to this simulation. Developing a realistic simulation like SIMPLER, which is known to closely correlate with real-world environments, is essential for meaningful transferability.
>
> ---
> **Q3: Window size ablation**
>
> For all robot manipulation videos, we have determined the window size depending on the fps of the video so that the next frame models 0.6 seconds ahead from the current frame. For human manipulation videos (Sthv2), we have set the next frame to be 2.4 seconds ahead since we qualitatively observed that many of the frames of the human videos contain much less dynamic actions compared to robot videos. (However, we think that filtering these frames could make the window size the same as robot videos, which we leave as future work.). We have added an ablation experiment on the window size for robot videos (Bridgev2) by evaluating on SIMPLER in Figure 15a of Appendix F.
> The results show that LAPA is quite robust to different window sizes. However, if the window size is extremely large, performance degradation is observed. This is expected since our quantization model is relatively small (300M parameters), it faces difficulties modeling latent information when the visual deltas are significant.

---

> > ### Author Response · Authors · 2024-11-24
> > **Comment by authors**
> >
> > As the discussion deadline approaches, the authors kindly encourage the reviewer to review the responses. We have thoughtfully revised the manuscript based on the valuable suggestions and would greatly appreciate any further feedback!

---

> > ### Comment · Reviewer_D2cm · 2024-11-26
> >
> > Thank you for addressing my concerns. I believe that the overall clarity of the paper improved.
> >
> > I still think the overall focus of the experiments could be improved. Most experiments use robot demonstrations, which shows that the method can effectively learn latent actions from robot demonstrations. However, this work's main contribution and claim is a method that can train policies from internet-scale videos without action labels. The experiments only support these claims to some extent.
> > The additional scaling experiments are promising but must be extended further to fully back the scaling claims, especially for human-robot domain transfer. Therefore, I will not raise my score.
> >
> > That said, I think the proposed method is convincing, and the SSv2 results suggest that the method works. In my opinion, this paper presents an important contribution to the robot learning community, and I still think it should be accepted.

---

> > > ### Author Response · Authors · 2024-12-03
> > > **Comment by authors**
> > >
> > > Thank you for your comment and suggestion. Let us know if you have further questions.

---

### Official Review · Reviewer_jNBu · 2024-11-02

**Soundness:** 3
**Presentation:** 4
**Contribution:** 3
**Rating:** 8
**Confidence:** 4

**Summary:**

This paper proposes a method to pretrain vision-language-action (VLA) models for robotic manipulation using video data without robot action labels. Starting with actionless videos, they first train a VQVAE to infer latent actions between consecutive frames. They then train a VLM to predict these latent actions based on the current image and the language instruction. Finally, they finetune this model using robot trajectories with action labels for the target tasks. Their approach is evaluated in simulated and real-world multi-task experiment settings, including pretraining on actionless robot videos followed by finetuning on  robot trajectories of different tasks and pretraining on human videos from the Something-Something-v2 dataset.

**Strengths:**

- Interesting approach: The proposed approach is both simple and practical, potentially easier to implement than the baselines considered in the experimental section. Pretraining VLAs on actionless data, especially human videos, is particularly relevant, and the use of inferred latent actions is a sensible solution.
- Good experimental results: Through extensive comparative and ablation studies in both simulation and real-world robot settings, the authors clearly demonstrate the effectiveness of their proposed method. The experiments involving pretraining on human videos are impressive.
- Clarity: The paper is well-written, clear, and easy to follow.

**Weaknesses:**

Since the latent actions are not directly used for downstream control and the model is finetuned on robot action labels, it’s unclear whether the performance gains come from leveraging temporal information/action priors in videos or simply from pretraining on data (robot trajectories/SSv2) that more closely aligns with the finetuning robot data compared to the base VLM’s original training data. Would a pretraining task without temporal information—such as image captioning— achieve similar results? For the same reason, it remains unclear if this approach can scale effectively to internet-scale video datasets, in particular beyond manipulation videos, as claimed by the authors.

**Questions:**

- Where do you think the performance gains by pretraining on videos come from (see above) ?
- Have you tried pretraining on human video datasets other than SSv2, such as EpicKitchens, Ego4d, or even datasets less focused on manipulation ?

---

> ### Author Response · Authors · 2024-11-18
> **Rebuttal by authors**
>
> We thank the reviewer jNBu for the helpful suggestions.
>
> ---
> **W1: Effect of Temporal Information / Action Priors**
>
> Regarding the point about performance gains potentially resulting from pretraining on in-domain data, recent work [1] challenges the assumption that pretraining on large object-interaction datasets—such as first-person videos of humans performing diverse tasks—automatically enhances performance in robotic applications. This is likely due to the absence of temporal information or action priors, focusing instead on matching the data distribution alone.
>
> [1] Burns, K., Witzel, Z., Hamid, J. I., Yu, T., Finn, C., & Hausman, K. (2023). What Makes Pre-Trained Visual Representations Successful for Robust Manipulation?. arXiv preprint arXiv:2312.12444.
>
> ---
> **Q1: Pretraining on other human video datasets**
>
> We believe this is an interesting direction for future work. We did not experiment with other video datasets, as they tend to be noisier than SSV2 (notably, we applied no filtering to the SSV2 dataset). However, with effective algorithms to filter out noisy video instances, the approach could show promising results on datasets like Epic Kitchens or Ego4D. As illustrated in Figure 14, LAPA not only captures agent-centric actions but also visual changes in the environment, such as camera motions, suggesting it can learn meaningful latent actions even from non-manipulation videos.

---

> > ### Comment · Reviewer_jNBu · 2024-11-22
> >
> > Thank you for your response.
> >
> > It seems that [1] focuses on pretrained image representations for manipulation, whereas your approach centers on pretraining the language model. Could you clarify your answer?
> >
> > Additionally, could you add a discussion about [2]? I missed this reference during my initial review, but it shares similarities with your work (pretraining on actionless video, VQ-VAE), so it would be better to include it in the paper if possible.
> >
> > [2] Learning an Actionable Discrete Diffusion Policy via Large-Scale Actionless Video Pre-Training, He et al., NeurIPS 2024

---

> > > ### Author Response · Authors · 2024-11-24
> > > **Response by authors**
> > >
> > > Thank you for your response.
> > >
> > > ---
> > >
> > > Yes, it is true that [1] focuses on pretrained image representations for manipulation. For the comparison between LAPA and a baseline model using the same data and backbone model (a pretrained VLM), we compare it with VPT, which is pretrained on the same data and backbone model but uses a different method to learn action priors, as described in Sections 4.3 and 4.4. The results show that LAPA achieves a better action prior compared to VPT, which also learns from the same pretraining data. Although pretraining on data more closely related to the downstream task could also improve downstream task performance, we believe that the effectiveness of LAPA stems from learning better action priors rather than solely from pretraining on task-related data.
> > >
> > > ---
> > > Regarding the missing reference, we have cited the suggested work in our revised version. Thank you for your suggestion. We believe VPDD shares similarities with our work in that it also learns discrete latent representations from actionless videos for pretraining. However, unlike LAPA, which learns latent 'actions' from visual deltas, VPDD discretizes the whole video to obtain a representation for the video itself.

---

### Official Review · Reviewer_5VGj · 2024-11-03

**Soundness:** 4
**Presentation:** 3
**Contribution:** 3
**Rating:** 6
**Confidence:** 4

**Summary:**

This paper proposes a novel pretraining method for VLAs. It first utilizes VQ-VAE to learn a encoder which encode the image into discrete token, and the decoder decode the next frame to learn the dyanmics of the adjacent frames. The discrete tokens could be treated as action tokens and replace the action labels in the human / robot rollouts.
The authors conduct  successive experiments to demonstrate the model's performance over in-domain scenes, cross-env scenes and cross embodiment scenairos to demonstrate the effectiveness of the proposed framework.

**Strengths:**

The innovative approach of using VQ-VAE to encode image dynamics into latent space and replacing labeled actions with these encoded tokens is particularly intriguing. This method holds significant importance for the research community, given the high costs associated with data collection for action labeling.

The experimental validation is comprehensive, with strong results obtained from both simulation environments and real-world settings, underscoring the reliability of the model.

The analysis is comprehensive, including the performance of in-domain, cross-environment and cross-embodiment setting.

The paper is well-structured and presented, making it easy for readers to follow the rationale and outcomes of the research.

**Weaknesses:**

The pretraining and finetuning setups in experiment section is a little confusing. For example, how is ActionVLA pretrained with action labels while there does not exist action labels in in something V2.

The utilized finetuning recipe of other baselines is not demonstrated in detail, which makes me concer the fairness of the comparison. I hope the authors could add detailed information in the appendix.

All the experiments in simulators are trained with only few trajectories, especially in Bridge V2, only 100 trajectories are utilized. I am wondering that if the model could learn multiple language conditioned tasks (like above 30 categories with diverse instruction) well. I suggests that tha authors add more experiments in a new simulator with more task catigories, and further train with RT-X data mixup and test the model performance in Simpler with Bridge and Google Robot.

**Questions:**

See weakness.

---

> ### Author Response · Authors · 2024-11-18
> **Rebuttal by authors**
>
> We thank Reviewer 5VGj for the helpful feedback.
>
> ---
> **W1: Pretraining and Finetuning setups**
>
> We trained ActionVLA only on robot manipulation videos that already have access to ground-truth action labels (e.g. Bridgev2). However, as you have pointed out, it is infeasible to do ActionVLA pretraining with human manipulation videos. Therefore, we do not have ActionVLA baselines in Section 4.6.
>
> ---
> **W2: Baseline Details**
>
> We have provided the baseline model details in Appendix C. Let us know if you need more details on the baseline setup.
>
> ---
> **W3: Multiple Language Conditioned Tasks**
>
> Thank you for the suggestion. In our simulation experiments, we used the Language Table dataset to evaluate diverse language conditioning capabilities and the SIMPLER dataset to assess higher DoF robot manipulation. We agree that while the Language Table dataset includes five categories with varied instructions, its diversity remains limited. Although we attempted to test broader language conditioning in SIMPLER, adding new tasks and assets proved challenging. We hope that future simulation benchmarks will provide more extensive language conditioning evaluations AND support higher action dimensions (beyond 2 DoF) to better assess these capabilities.

---

> > ### Author Response · Authors · 2024-11-24
> > **Comment by authors**
> >
> > As the discussion deadline approaches, the authors kindly encourage the reviewer to review the responses. We have thoughtfully revised the manuscript based on the valuable suggestions and would greatly appreciate any further feedback!

---

### Official Review · Reviewer_CUQQ · 2024-11-03

**Soundness:** 4
**Presentation:** 3
**Contribution:** 4
**Rating:** 6
**Confidence:** 4

**Summary:**

This paper proposes an unsupervised pre-training method, LAPA, for vision-language-action (VLA) models that eliminates the need for action labels. LAPA uses a VQ-VAE structure to learn quantized action latents from frame differences. Subsequently, the authors train a VLA model to predict these quantized latents based on frames and language instructions. This model can then be fine-tuned on small-scale robot manipulation datasets. Through experiments on both simulated and real-world datasets, LAPA demonstrates improved performance over selected baseline models to indicate its generalization ability.

**Strengths:**

Originality:
- The proposed method removes the need of action labels for pre-training VLA models which significantly increase the data availability.
- Training VQ to predict delta between frames is a simple and scalable way of learning coarse latent action.
- A significant performance improvement compared to SoTA (OpenVLA) model under various scenarios and relatively small performance gap between the upper bound case (ActionVLA) and LAPA.

Quality:
- The proposed method is technically sound.
- Extensive experiments are conducted to evaluate LAPA performance under various scenarios.

Clarity:
- The paper is overall well-written and easy to follow.

Significance:
- LAPA provides a way of utilizing large amounts of videos without action labels with huge embodiment distribution shifts, which is a significant contribution to scalable robot learning.

**Weaknesses:**

- Lack of Experiments on Sequence Length in VQ Stage: There is a lack of experiments illustrating the effect of different sequence lengths during the VQ stage. It seems arbitrary that the latent code length is set to 4 (line 433-434), and for the language table dataset (line 933), the sequence length is set to 1. A discussion on the rationale behind these choices is missing. Incorporating experiments on various sequence lengths could help assess LAPA’s flexibility and robustness.

- Limited Ability to Capture Complex Movements: Learning frame differences may only capture simple movement information. In visualizations of the learned latents, it appears that LAPA’s latent code primarily contains embodiment or camera movement information. Detailed results (Tables 13, 14, and 15) also indicate that LAPA performs better on coarse movement tasks like knocking and covering but struggles with finer actions like grasping or picking objects. This suggests that frame-difference learning might not be ideal for learning action latents. Including experiments with alternative approaches could provide a clearer evaluation.

- Impact of Smaller Action Space: The smaller action space in LAPA compared to OpenVLA (256 bins for each action dimension in OpenVLA vs. a relatively smaller space in LAPA) may account for the observed performance improvement, rather than frame-difference learning alone. Reducing OpenVLA’s latent space for fairer comparison could better clarify the contributions of each part of LAPA.

- Minor Wording Issue: The phrase “actionless video” on line 83 may be misleading, as it implies the absence of any action. Consider rephrasing for clarity.

**Questions:**

- Need clarification for pretraining stage: In table 1, authors provide details about the dataset for both pre-train and finetune phase, however, it remains unclear about the latent action quantization. My understanding is that latent action quantization is part of the pretraining so both latent quantization  (section 3.1) and latent pretraining (section 3.2) use the same dataset. Clarification of the dataset usage could provide a better understanding of the experiment set up.

- In most of the experiments, LAPA is trained with a single large scale dataset for fair comparison with other models like OpenVLA. Since LAPA does not require any action label during pre-training, it would be interesting to see having all three datasets used together can provide benefit of downstream performance boost to further backup the scalable claim.

---

> ### Author Response · Authors · 2024-11-18
> **Rebuttal by authors**
>
> We thank Reviewer CUQQ for the helpful comments.
>
> ---
> **W1: Sequence Length Ablation**
>
> Thank you for the suggestion. We have added latent action sequence ablation results for the SIMPLER and Language Table datasets in Figures 5 and 16. For SIMPLER, we observe that increasing the sequence length is crucial for performance improvement. In contrast, for the Language Table dataset, while a longer sequence length is beneficial, increasing the vocabulary size proves to be more effective, even with a smaller action space.
>
> ---
> **W2: Limited Ability to Capture Complex Movements**
>
> We agree that frame-difference learning might not be ideal for learning complex movements with the current scale. We think that there are two potential solutions to this: 1) increasing the latent action size (vocabulary and sequence length) to capture for fine-grained visual deltas or 2) attaching a small action expert for better fine-grained control, following [1]. Through this architecture, the VLA can plan coarse-grained actions from its language-conditioning capability while the action expert can generate fine-grained actions with high-level frequency.
>
> [1] π0 : A Vision-Language-Action Flow Model for General Robot Control (Black et al, 2024)
>
> ---
> **W3: Smaller Action Space**
>
> Thank you for your suggestion. We believe the advantage of a smaller action space becomes more apparent when using smaller pretraining datasets (e.g., Bridgev2 with 50–60K trajectories) or datasets with simpler distributions (e.g., Language Table), rather than larger real-world datasets (e.g., Open-X Embodiment with 970K trajectories). Training ActionVLA on the Open-X Embodiment dataset is left as future work due to computational constraints.
>
> ---
> **W4: Wording**
>
> Thank you for your suggestion. We have rephrased into “policies that are pretrained without using ground truth action labels”.
>
> ---
> **Q1: Clarification of Latent Action Quantization**
>
> Yes, your understanding is correct. For all experiments, we keep the dataset of latent action quantization and latent pretraining identical. We have added the dataset details regarding latent action quantization in the revised version.
>
> ---
> **Q2: Multiple Dataset Training**
>
> We agree that combining all datasets, including both robot and human videos, is a promising direction for future work. Although we have not pursued this due to limited computational resources, training on such a unified dataset would be a natural next step.

---

> > ### Author Response · Authors · 2024-11-24
> > **Comment by authors**
> >
> > As the discussion deadline approaches, the authors kindly encourage the reviewer to review the responses. We have thoughtfully revised the manuscript based on the valuable suggestions and would greatly appreciate any further feedback!

---

> > ### Comment · Reviewer_CUQQ · 2024-11-25
> >
> > Thank you for your response.
> >
> > The authors’ response has clarified some of my questions. However, I have a few additional questions and comments outlined below:
> > - W1 & W3: Sequence Length Ablation and Smaller Action Space
> > From Fig. 16, I can observe performance improvement across a wide range of action vocabulary sizes (from 8 to 1024) for the Language Table dataset. However, in Fig. 5, the action vocabulary for the SIMPLER dataset is considerably smaller, ranging only from 2 to 8. This discrepancy in vocabulary size between the two datasets is substantial. Could you clarify why this discrepancy exists? Does this suggest that, even with the current dataset scale, LAPA requires careful tuning of the action space to achieve good performance? If so, might this limit the scalability of the approach or its applicability to combined or larger datasets?
> > - W2 & Q2: Limited Ability to Capture Complex Movements & Data Scaling
> > As reviewers D2cm and 2CRD also pointed out, data scaling is key to the value of LAPA. Since LAPA pretraining relies on a relatively simple supervision signal (frame delta), conducting experiments that explore the data scaling properties would significantly enhance readers’ understanding of the effectiveness of using latent codes of frame delta as a substitute for action labels. Including such scaling experiments could have a positive impact on the overall strength of the paper.

---

> > > ### Author Response · Authors · 2024-11-25
> > > **Response by authors**
> > >
> > > Thank you for your comment.
> > >
> > > ---
> > > **W1 & W3:**
> > >
> > > The discrepancy in vocabulary size between the two datasets arises from the observation that “increasing the latent action sequence length is less effective compared to increasing the vocabulary for Language Table”. In the latent action vocabulary ablation for SIMPLER (Fig. 5(d)), the latent action sequence length is set to 4 $(2^4, 4^4, 8^4)$. In contrast, for the Language Table dataset (Fig. 16(b)), the sequence length is fixed at 1 $(8^1, 256^1, 1024^1)$, as increasing the sequence length provided less benefit for this dataset compared to SIMPLER.
> > >
> > > We hypothesize that this difference stems from the visual complexity disparity between the two datasets. The latent action sequence length is determined by the capacity of a CNN network, and the Language Table dataset, being visually simpler, benefits more from vocabulary size increases than sequence length adjustments.
> > >
> > > Importantly, for all other experiments (Bridgev2, Open-X, Something Something V2), a fixed configuration of a vocabulary size of 8 and sequence length of 4 was used. This consistency highlights the universality of LAPA's design without requiring careful tuning, except for datasets like the Language Table, where visual simplicity warrants a different configuration.
> > >
> > > ---
> > > **W2 & Q2:**
> > >
> > > We agree that data scaling is crucial to the value of LAPA. Figure 5(b) presents the data scaling results for Bridgev2 during pretraining. For human videos, we also ran an additional variation of the Something Something V2 dataset using 10% of its original size of 220K videos. Results show that scaling the human video datasets boosts the performance for SIMPLER benchmark not only for the final success (50.0 → 52.1) for all subtasks (Grasping: 56.2 → 66.7 / Moving: 62.5 → 72.9). We leave exploring scaling law for human videos more extensively or future work, since showing scaling law requires intensive computational resources to do different ablations of model size, data size, and computational resources. However, these preliminary findings underscore the potential of scaling human video datasets to further enhance performance.

---

### Public Comment · ~Inseop_Chung1 · 2024-11-29
**Two Questions about LAPA**

Hello, first of all, thank you for the great work you provided!
I think it is a remarkable work!

I have two questions about LAPA.

1. What is the advantage of learning the action in the latent space ?
    Why not directly learn the action in the true action space?
    For example, why not make the IDM directly output actual robot actions?
    I would highly appreciate if you can provide some of the prior works that show utilizing latent space for action learning is advantageous.

2. Why is LAPA robust to unseen tasks ?
    What part of LAPA makes it generalizable to unseen scenarios (unseen objects, unseen combinations of seen objects)?

Thank you

---

> ### Author Response · Authors · 2024-12-03
> **Reply by authors**
>
> Thank you for your interest.
> 1. The advantage of learning robot actions in latent space has also been explored in previous work [1]. However, unlike our work, [1] uses ground truth action labels to learn latent actions. Also, one of our baselines in the paper, VPT, learns an IDM to generate robot actions. However, as shown in the results of the paper, VPT is not robust to the distribution shift between pretraining and finetuning compared to LAPA.
>
> 2. As discussed in the main paper, we hypothesize that LAPA is robust to unseen tasks due to leveraging a shared representation output space across different datasets and embodiments, enhancing positive transfer when trained on multiple datasets and embodiments.
>
> [1] Lee et al, Behavior Generation with Latent Actions (2024)

---

### Meta-Review · Area_Chair_kXfx · 2024-12-21

**Metareview:**

After careful consideration of the reviews, the subsequent author-reviewer discussions, and my own reading of the paper, I recommend accepting this submission. The paper presents an approach to pretraining Vision-Language-Action (VLA) models without requiring ground-truth robot action labels, and demonstrates potential for scaling robot learning through the use of widely available video data.

The paper's primary technical contribution, LAPA, addresses a challenge in robot learning by enabling the use of unlabeled video data for pretraining. The method combines VQ-VAE-based action quantization with a latent VLA model in a technically sound manner. The authors provide comprehensive experimental validation across multiple domains, including both simulated and real-world environments, which demonstrated the method's effectiveness and generalization capabilities.

The review process generated substantial discussion, particularly regarding the method's scalability and experimental validation. While reviewer 2CRD raised important concerns about the need for more extensive scaling experiments with the Something-Something V2 dataset, their expectation for GPT3-style scaling laws may be a bit overly demanding for an academic publication. The other reviewers (CUQQ, 5VGj, jNBu, D2cm, and 7bYx) provided balanced assessments that recognized both the method's current limitations and its contributions.

Several reviewers noted the paper's thorough experimental evaluation, including ablation studies and comparisons with state-of-the-art baselines. The authors' responses during the discussion period were constructive and thorough, addressing concerns about implementation details, clarifying methodological choices, and providing additional experimental results where feasible within computational constraints.

The decision to accept is based on the paper's technical contribution, thorough experimental validation within reasonable academic constraints, and potential impact on the field of robot learning. While more extensive scaling experiments would be valuable, their absence does not diminish the paper's immediate contributions to the community.

**Additional Comments On Reviewer Discussion:**

During the rebuttal period, there was extensive and constructive discussion between the authors and reviewers regarding several key aspects of the paper. The most substantive debate centered on the paper's empirical validation and scalability claims.

Reviewer 2CRD initially raised significant concerns about data consistency and the paper's claims regarding scalability to internet-scale data. The authors clarified that certain inconsistencies were limited to appendix tables and did not affect the main results. The discussion evolved to focus on the need for more comprehensive scaling experiments with the Something-Something V2 dataset. While the authors provided additional results, the reviewer maintained that more extensive scaling experiments were necessary to validate the paper's claims about internet-scale potential.

Several reviewers, including CUQQ and D2cm, requested clarification about the paper's technical details and methodology. The authors responded by providing additional ablation studies on window size selection and detailed explanations of their architectural choices. These responses effectively addressed the technical concerns, leading these reviewers to express satisfaction with the clarifications provided.

Reviewer 7bYx questioned the method's performance across different domains and tasks, particularly in comparison to OpenVLA. The authors provided additional experimental results and detailed explanations of how pretraining data distribution affects performance across different tasks. This transparent discussion of limitations and their causes strengthened the paper's credibility.

In weighing these discussions, the area chair found that while the concern about extensive scaling experiments raised by reviewer 2CRD is valid, the expectation for GPT3-style scaling laws may be less reasonable given typical academic resource constraints. The authors' demonstration of meaningful improvements with scaled data, combined with their thorough technical validation and transparent discussion of limitations, provides sufficient evidence of the method's potential while acknowledging areas for future work.

---

### Decision · Program_Chairs · 2025-01-22

Accept (Poster)